# Reasoning as Representation: Rethinking Visual Reinforcement Learning in Image Quality Assessment

**Shijie Zhao**[1,†], **Xuanyu Zhang**[1,2,†], **Weiqi Li**[1,2], **Junlin Li**[1], **Li Zhang**[1],
**Tianfan Xue**[3], **Jian Zhang**[2]
[1]ByteDance Inc., [2]Peking University, [3]The Chinese University of Hong Kong

## Abstract

Reasoning-based image quality assessment (IQA) models trained through reinforcement learning (RL) exhibit exceptional generalization, yet the underlying mechanisms and critical factors driving this capability remain underexplored in current research. Moreover, despite their superior performance, these models incur inference energy usage and latency orders of magnitude higher than their earlier counterparts, restricting their deployment in specific scenarios. Through extensive experiments, this paper verifies and elaborates that through RL training, MLLMs leverage their reasoning capability to convert redundant visual representations into compact, cross-domain aligned text representations. This conversion is precisely the source of the generalization exhibited by these reasoning-based IQA models. Building on this fundamental insight, we propose a novel algorithm, RALI, which employs contrastive learning to directly align images with these generalizable text representations learned by RL. This approach eliminates the reliance on reasoning processes and even obviates the need to load an LLM during inference. For the quality scoring task, this framework achieves generalization performance comparable to reasoning-based models while requiring less than 5% of their model parameters and inference time. Code is available at RALI.

## 1 Introduction

Image Quality Assessment (IQA) is a fundamental task in the field of computer vision, with application scenarios covering two key dimensions. In natural scenarios, it supports photography selection and video platform quality monitoring (Sheikh, 2005; Lin et al., 2019; Fang et al., 2020; Wu et al., 2024b); In the field of generative algorithms, IQA serves as a core reward signal in the Reinforcement Learning from Human Feedback (RLHF) framework (Rombach et al., 2022; Dhariwal & Nichol, 2021; Wang et al., 2025; He et al., 2024), which is crucial for the training process of generative image and video models. Its performance directly affects the convergence efficiency and the effect of reinforcement learning strategies.

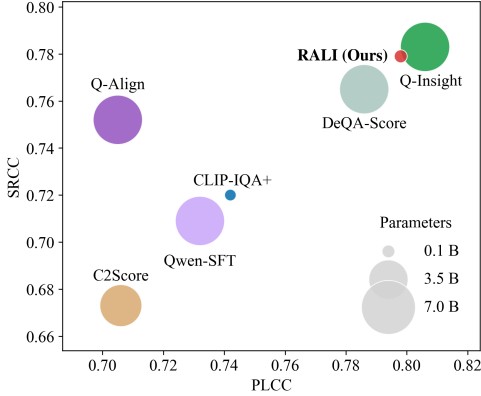

Figure 1: **Performance comparison among IQA methods in PLCC/SRCC and parameter numbers.** RALI uses only about 4% of Q-Insight's (Li et al. (2025)) parameters while achieving comparable accuracy.

With the development of multimodal large language models (MLLMs), a series of innovative methods have emerged in the IQA field. Q-Align (Wu et al., 2024b) and DeQA (You et al., 2025) enable MLLMs to directly output image quality scores through supervised fine-tuning (SFT). Descriptive algorithms such as DepictQA (You et al., 2024b) focus on the text representation of image quality. Recently, studies represented by Q-Insight (Li et al., 2025; Zhang et al., 2025a) and VisualQuality-R1 (Liu et al., 2025c) have introduced visual reinforcement

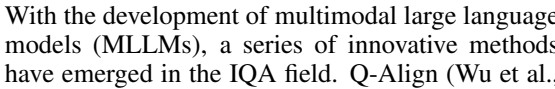

---

[1]†These authors contributed equally to this work.

learning (RL) into IQA tasks by outputting quality reasoning text during reasoning and scores afterward. Their generalization in image quality prediction is significantly superior to previous SFT methods. Despite their strong performance, two challenges remain. First, the principle behind their generalization improvement lacks systematic analysis. Notably, while studies have explored RL generalization in other fields (Chu et al., 2025; Liu et al., 2025c; Pan et al., 2025; Xu et al., 2025), the unique complexity of visual characteristics and the subjectivity of quality evaluation in IQA tasks render direct transfer of these findings difficult. Second, stepwise reasoning incurs high latency and loading overhead, limiting deployment in online RL, mobile, and real-time settings. This raises two critical questions: *How is generalization related to reasoning in IQA, and is it essential?* To address the above questions, this paper focuses on the source of generalization of RL-based IQA models (e.g., Q-Insight (Li et al., 2025)).

Turning to the first question—*How is generalization related to reasoning in IQA?* Generalization has been a topic of discussion for IQA tasks, as individual datasets are typically small-scale and there is a pronounced domain gap between them due to their varying distributions in image quality and label annotations (You et al., 2025). Thus, using high-dimensional visual representations to predict scores tends to lead to overfitting. However, experimental verification reveals a key finding. For reasoning-based models like Q-Insight, the dependence in their scoring process has changed almost entirely. Instead of relying on lengthy visual tokens, it now depends on concise and compact quality reasoning text tokens. The core mechanism is that RL methods (e.g., GRPO (Shao et al., 2024; Guo et al., 2025)) enable MLLM to acquire a dimensionality reduction strategy: reasoning, which manifests itself as mapping input images to quality reasoning text to form IQA representations. Specifically, previous MLLMs typically predict image quality through visual representations (more than 1000 tokens), whereas reasoning-based models rely primarily on the textual representations (less than 100 tokens), resulting in a compression of more than 10 times. Furthermore, we further demonstrate that the text representations can mitigate domain discrepancies. Meanwhile, the reasoning process itself, that is, the conversion of images to quality reasoning text, exhibits weak correlation with specific datasets and can maintain stable alignment across different domains. Together, these factors explain the generalization of reasoning-based IQA models. We further validate the generalization of image quality reasoning text by proposing a novel **R**easoning-**A**ligned **C**ross-domain **T**raining (**RACT**) framework. This approach addresses dataset distribution issues in image quality assessment tasks, enabling effective cross-domain training in misaligned data scenarios.

Turning to the second question: *Is reasoning essential?* From our prior discussion, we know that LLM reasoning maps images to quality reasoning text to achieve generalization, but contrastive learning methods, such as CLIP (Radford et al. (2021)) (which maps text and images to a shared embedding space), can also accomplish this mapping without the need for multistep reasoning, potentially offering a new pathway for generalization in IQA. Existing CLIP-based IQA methods (e.g., CLIP-IQA (Wang et al., 2023)) have similar attempts but two flaws: shallow alignment with general text (not quality-specific) and overly simplistic text corpora (e.g., only "good/bad photo" and lacking complex quality dimensions). Building on prior research into Visual RL training mechanisms and reasoning processes, we have addressed these limitations and proposed a **R**easoning-**A**ligned **L**ightweight IQA (**RALI**) framework. RALI consists of three key steps: First, we acquire image-text-score data triplets via reinforcement learning; second, we align images with quality reasoning text through CLIP-based contrastive learning; finally, for the textual score space, we leverage description-score pairs to define a higher-dimensional, more sophisticated score mapping space. For score prediction, we directly map images to the pre-constructed image quality text space (without retaining the reasoning process) and accomplish scoring via intra-space similarity calculation. As shown in Fig. 1, RALI uses only 4% of Q-Insight's parameter while achieving comparable scoring accuracy. Extensive experiments demonstrate that RALI achieves generalization on par with RL-based MLLMs, outperforms all other methods, and eliminates both the reasoning process and the deployment of LLMs, reducing the inference time and the memory by more than 95%.

In summary, this paper shows that the generalization of reasoning IQA model is rooted in the compression of visual information into textual representations, and this finding is further corroborated by RACT. Additionally, we prove that an equivalent level of generalization can be realized through RALI, a framework that does not incorporate reasoning process or depend on LLMs.

## 2 RELATED WORKS

**Image Quality Assessment.** Classical research divides IQA into full reference and no reference settings. Full reference methods (Wang et al. (2004); Sheikh & Bovik (2006); Zhang et al. (2011)) compare a distorted image with a pristine reference using traditional metrics such as SSIM (Wang et al. (2004)) as well as deep learning based metrics (Bosse et al. (2017); Cao et al. (2022); Ding et al. (2020; 2021); Ghildyal & Liu (2022); Prashnani et al. (2018)) like LPIPS ( Zhang et al. (2018)). No reference methods estimate perceptual quality without an explicit reference, evolving from hand-crafted natural scene statistics (Ma et al. (2017); Mittal et al. (2012a;b); Moorthy & Bovik (2010; 2011); Saad et al. (2012)) to models that learn strong quality priors from data (Kang et al. (2014); Ke et al. (2021); Liu et al. (2017); Pan et al. (2018); Su et al. (2020); Zheng et al. (2021); Zhu et al. (2020); Sun et al. (2022); Wang et al. (2023)).

**MLLM-Based IQA Methods.** Recent work employs multi-modal large language models (MLLMs) to assess image quality. Score-based approaches such as Q-Align (Wu et al. (2024b)) and DeQA-Score (You et al. (2025)) produce numerical ratings by leveraging the models' perception and world knowledge, yet they limit the intrinsic descriptive ability of MLLMs. Description-based approaches (Wu et al. (2025a; 2024a); You et al. (2024b;a); Wu et al. (2024c); Chen et al. (2024); Zhang et al. (2025c;d;b)) aim to deliver qualitative judgments with richer explanations and better interpretability, while relying on large volumes of textual annotations for supervised fine tuning (SFT). Very recently, visual reinforcement learning is introduced into IQA tasks (Li et al. (2025); Zhang et al. (2025a); Wu et al. (2025b)). These RL-based IQA methods can jointly output quality reasoning text and scores, and show superior generalization ability to SFT-based methods.

## 3 REVISITING REASONING-BASED MLLMs IN IQA

### 3.1 PRELIMINARIES

**Reinforcement Learning for Image Quality Assessment.** Reinforcement learning (RL) improves the reasoning ability of large language models through feedback driven refinement (Christiano et al. (2017); Silver et al. (2017); Shao et al. (2024); Yang et al. (2024); Ying et al. (2024); Hui et al. (2024); Zhang et al. (2024)). Recently, DeepSeek-R1 Zero (Guo et al. (2025)) introduces group relative policy optimization (GRPO) (Shao et al. (2024)), which strengthens reasoning using rule-based rewards and avoids heavily supervised fine-tuning. In the context of visual quality understanding, Q-Insight (Li et al. (2025)) firstly integrates GRPO by using quality scores to construct rule based rewards and by training two tasks jointly, score regression and degradation perception. For each image and task specific question, the policy generates groups of answers with explicit reasoning, task specific evaluators compute rewards for score regression and for degradation type and level, and a multi task GRPO update increases the likelihood of higher reward answers while a KL regularizer keeps the policy close to a fixed reference. The generalization of Q-Insight is significantly enhanced by training that incorporates reasoning, this critical factor is further elaborated with explanations and experiments in the Appendix A. In inference, Q-Insight outputs reasoning contents (between <think> and </think>) and then score results (between <answer> and </answer>), with the first being image quality reasoning text.

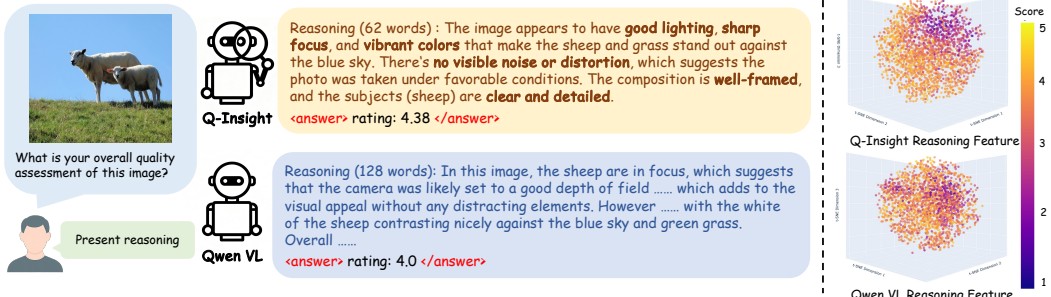

Figure 2: **Comparison of Reasoning Between Q-Insight and Qwen-VL on the IQA Task.** (left) Q-Insight's reasoning was more concise than Qwen-VL's and more correlated with image quality. (right) On the KonIQ (Hosu et al. (2020)) test set, CLIP features derived from Q-Insight's reasoning were more correlated with scores following t-SNE visualization. In this paper, we adopt the quality reasoning text between <think> and </think> as the reasoning contents for clarity.

## 3.2 Reasoning Mechanisms of MLLMs in Image Quality Scoring.

In this paper, we take Q-Insight as a case study for elaboration. We analyze the reasoning differences between Q-Insight and Qwen VL (Bai et al. (2025)) in the image quality scoring task, where the former is a model fine-tuned by reinforcement learning for image quality scoring based on Qwen VL , while the latter is a general purpose MLLM. Two critical observations emerged from our analysis. First, it is observed that Q-Insight generated more concise descriptions with less unrelated information. Second, we found that reinforcement learning training significantly improved the correlation between quality reasoning text and subjective quality scores. As shown in Figure 2, we extracted the features through CLIP (Radford et al. (2021)) from the reasoning outputs of Q-Insight and Qwen VL on the KonIQ testset and visualized these features using t-SNE (van der Maaten & Hinton (2008)).

To further uncover the relationship between reasoning and scoring, we visualized its attention heatmap during the generation of score tokens. As illustrated in Figure 3, when the model outputs score tokens, **95%** attention weights are assigned to the previously generated reasoning text tokens (excluding the fixed prompt).

Based on the analysis above, we conclude that through reinforcement learning, the reasoning model shifted its dependency on image quality scoring from visual tokens to reasoning text tokens, and its text tokens were more concise and more relevant to image quality.

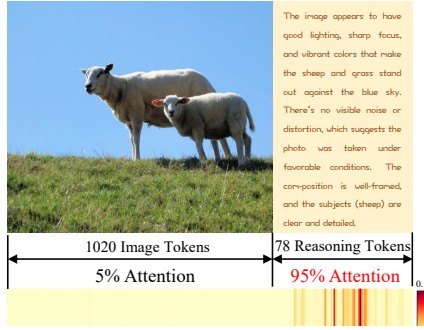

Figure 3: **Attention heatmap during score-token generation of Q-Insight.** It primarily attends to reasoning text tokens instead of visual tokens (95% vs. 5%).

## 3.3 Key to Generalization: Compressing Visuals into Text Representation

**Text is a More Compact and Domain-bridging IQA Representation.** It is well-established that at comparable levels of representational capacity, more compact one exhibits better generalization (Bengio et al. (2013)), and text exhibits this trait. A 512×384 image requires approximately 1,000 tokens when using visual representations to predict the quality score, while fewer than 100 tokens suffice with text representations. Furthermore, reasoning models leveraging text for image quality prediction yield in-domain decent performance, confirming representational capability of the text.

Text representations can mitigate the domain gap between different datasets. We conducted reinforcement learning training on KonIQ (Hosu et al. (2020)) and SPAQ (Fang et al. (2020)) separately and visualized the visual tokens and reasoning text tokens of the two datasets during the image quality assessment process using t-SNE. Figure 4 demonstrates that the feature distributions of the datasets display notable disparities in the visual representational space, and training on features with this prominent domain gap is likely to reduce generalizability. Conversely, this domain gap is alleviated in the textual representational space.

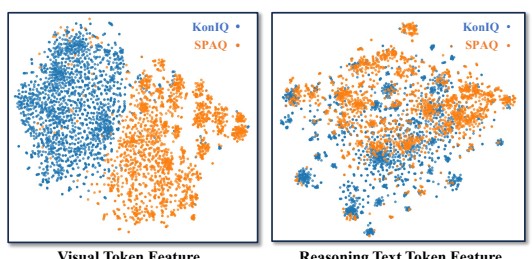

Figure 4: **t-SNE visualization** of visual tokens and reasoning text tokens from SPAQ and KonIQ.

**IQA Reasoning is a Generalizable Image-to-Text Compression.** We further demonstrate that the reasoning process itself exhibits strong generalization in IQA tasks, that is, the reasoning processes learned across different datasets are comparable and reduce the risk of overfitting.

We conducted an experiment in which we performed RL fine-tuning on the KonIQ (Hosu et al. (2020)) and KADID (Lin et al. (2019)) datasets separately, starting from a pre-trained Qwen VL model, while retaining only its LLM components, as these are responsible for the reasoning process. To isolate the effect of the Visual Encoder, we then paired these LLMs with the pre-trained Qwen Visual Encoder and evaluated them across multiple datasets. As shown in Table 1, the two LLM models exhibited little differences on out-of-domain datasets, CSIQ (Larson & Chandler (2010)) and LiveW (Ghadiyaram & Bovik (2015)), with PLCC difference within 0.01. However, for in-

Table 1: **PLCC/SRCC Comparison of Q-Insight trained on KonIQ and KADID using the Qwen Visual Encoder.** Out-of-domain results demonstrate that LLM and its reasoning processes trained across different datasets are highly consistent.

| Composition | | In-domain | | Out-of-domain | |
|---|---|---|---|---|---|
| Visual Encoder | LLM Trained on | KonIQ | KADID | CSIQ | LiveW |
| Qwen VL | KonIQ | 0.876 / 0.844 | 0.752 / 0.749 | 0.832 / 0.789 | 0.837 / 0.789 |
| Qwen VL | KADID | 0.810 / 0.749 | 0.841 / 0.841 | 0.839 / 0.790 | 0.832 / 0.791 |

domain evaluations, the scoring results for the respective datasets were higher, resulting from a closer alignment between scoring preferences and in-domain characteristics.

Now we can answer the question: ***How is generalization related to reasoning in IQA?*** In summary, reinforcement learning enables the model to acquire a highly generalizable compression from visual tokens to text tokens. With the strong representational capability and generalization of text tokens, the scoring process exhibits excellent generalization.

### 3.4 REASONING-ALIGNED CROSS-DOMAIN TRAINING FRAMEWORK

To further demonstrate quality reasoning text as an excellent representation of IQA, we propose that it could offer a new path to align datasets with varying distributions. Dataset variation from divergent distributions is one of the key challenges of IQA. To address this, ranking-based NR-IQA models have been introduced and adopted in prior MLLM-based IQA works (e.g., Compare2Score (Zhu et al. (2024)), DeQA (You et al. (2025)), VisualQuality-R1 (Wu et al. (2025b))). However, this challenge worsens in RL, as cross-dataset reward acquisition issues impede learning optimal reasoning paths. In particular, Q-Insight shows severe convergence problem on mixed datasets, while VisualQuality-R1's ranking-based training mitigates this, it still degrades with extensive mixed samples—e.g., its PLCC on KonIQ decreased by 0.024 compared to standalone training. These observations will be presented in the subsequent experimental section.

Based on the aforementioned analysis of the reasoning model scoring scheme and its generalization, we design the **R**easoning-**A**ligned **C**ross-Domain **T**raining (RACT) framework to enable co-training on multiple IQA datasets. First, we conduct independent reinforcement learning training on each IQA dataset (single-domain RL training). Second, we perform label alignment by leveraging the reasoning module to generate image quality reasoning text for each dataset, thereby forming unified cross-dataset labels in the form of image-text pairs. Third, we perform cross-domain SFT on the single-domain RL-trained model using aligned image-text pairs. For the schematic diagram, please refer to Figure A.1 in the appendix. Given that both the reasoning process (from visual tokens to text tokens) and the description outputs are cross-domain aligned, the aforementioned training can be implemented across domains. This training process primarily aims to adapt the visual encoder to images of varying quality and scenes, enabling it to effectively convert them into visual tokens. Furthermore, we only introduce score information from a single dataset during training, as multi-domain scores impair convergence.

## 4 REASONING-ALIGNED LIGHTWEIGHT IQA FRAMEWORK

We have revealed that reasoning is the key to generalization, but ***is reasoning essential***? To answer this question, we design a **R**easoning-**A**ligned **L**ightweight **I**QA framework, dubbed **RALI**. It aligns the reasoning description text produced by visual RL with the Visual Encoder, enabling it to achieve performance close to RL-based IQA methods while offering strong advantages in speed and memory usage. Specifically, as illustrated in Figure 5, our approach follows the basic pipeline "Visual token → Text token → Score", and consists of three steps: contrastive alignment, feature compression, and scoring definition.

**Contrastive Alignment.** First, we use a pre-trained reasoning-based IQA model Q-Insight to generate reasoning texts on its training set $\mathbb{C}$ such as (Hosu et al., 2020). Concretely, we encourage the scoring model to assign quality scores and extract the reasoning text from <think> and </think>, forming image–text–score triplets $\{\mathbf{I}, \mathbf{T}, s\}$. Notably, for the same input image, we use different seeds to enrich the diversity of the generated quality reasoning text. We then finetune a CLIP (Radford et al., 2021) vision encoder with an image–text contrastive learning loss (Radford et al., 2021) so that it aligns with the underlying quality reasoning text space. To be noted, we freeze the text encoder and only train the image encoder during the process.

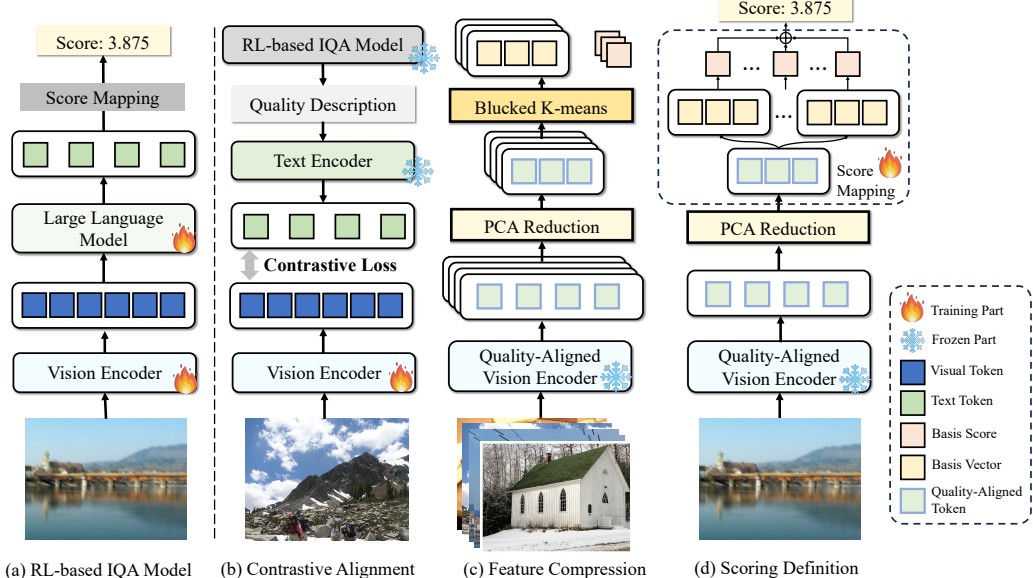

Figure 5: **Illustration of the proposed reasoning-aligned lightweight IQA (RALI) framework.** (a) presents the components and functions of the RL-based IQA model. (b)–(d) jointly constitute our lightweight RALI scheme, including contrastive learning with quality reasoning text, feature compression, and score definition. The model's inference pipeline is identical to (d).

**Feature Compression.** With the finetuned vision encoder $\mathcal{E}_{align}$, we convert the $L$ images in the training set $\mathbb{C}$ into $L$ embeddings $\mathbf{E} \in \mathbb{R}^D$, here $D$ is set to 768. Although high-dimensional visual embeddings can fit the feature space well, they may harm out-of-distribution generalization (Bengio et al., 2013), so we compress the visual tokens and reduce the visual space. We first apply the principal component analysis (PCA) to further reduce the embeddings from $\mathbb{R}^D \rightarrow \mathbb{R}^M$, here $M = 512$, producing the compressed embeddings $\hat{\mathbf{E}}$ and projection matrix $\mathbf{U} \in \mathbb{R}^{D \times M}$. This process further reduces dimensionality and filters out quality-irrelevant information from raw features. Then, to further reduce the number of embeddings and ensure that the retained embeddings correspond to a relatively dispersed score distribution, we partition the score range $[1, 5]$ into $N$ buckets and define $\mathcal{I}_n$ as the index set of samples whose scores fall into the $n$-th bucket, i.e., $\mathcal{I}_n = \{m : s_m \in \text{bucket } n\}$, where $s_m$ denotes the ground-truth score of sample $m$. For each bucket $n \in \{1, \dots, N\}$, we perform $k$-means with $k_n$ clusters indexed by $j \in \{1, \dots, k_n\}$.

$$r_{mj}^{(n)} = \mathbf{1}\Big[ j = \text{argmin}_{j' \in \{1,\dots,k_n\}} \big\| \hat{\mathbf{E}}_m - \boldsymbol{\mu}_{nj'} \big\|_2^2 \Big], \quad m \in \mathcal{I}_n; \tag{1}$$

$$\boldsymbol{\mu}_{nj} = \sum_{m \in \mathcal{I}_n} r_{mj}^{(n)} \hat{\mathbf{E}}_m \Big/ \sum_{m \in \mathcal{I}_n} r_{mj}^{(n)}, \quad f_{nj} = \sum_{m \in \mathcal{I}_n} r_{mj}^{(n)} s_m \Big/ \sum_{m \in \mathcal{I}_n} r_{mj}^{(n)}, \tag{2}$$

where the vector $\hat{\mathbf{E}}_m \in \mathbb{R}^M$ denotes the compressed embedding of sample $m$, the binary assignment variable $r_{mj}^{(n)} \in \{0, 1\}$ denotes whether $\hat{\mathbf{E}}_m$ is assigned to cluster $j$ in bucket $n$, determined by the indicator $\mathbf{1}[\cdot]$ and the nearest-centroid rule. The cluster centroid in bucket $n$, cluster $j$, is $\boldsymbol{\mu}_{nj} \in \mathbb{R}^M$, computed as the mean of assigned embeddings, and its representative score is $f_{nj}$, the mean of their scores. $\| \cdot \|_2$ denotes the Euclidean norm. Finally, our bucketed $k$-means yields a compact feature space with $K$ embeddings and scores aggregated over all buckets $\{\boldsymbol{\mu}_i, f_i\}_{i=1}^K$, where $K = \sum_{n=1}^N k_n$ and we flatten $(n, j)$ to a global index $i$ for simplicity, here $K = 250$, $N = 240$.

**Scoring Definition.** After obtaining the compact representation space, we further finetune these basic vectors $\boldsymbol{\mu}_i$ and their scores $f_i$ using image–score pairs $\{\mathbf{I}, s\}$ to better match human scoring preference. Given an input image $\mathbf{I}$, we use the aligned encoder $\mathcal{E}_{align}$ to compute its image embedding and project it to $\mathbb{R}^M$ via the PCA matrix $\mathbf{U}$. Then, we compute cosine similarities with the $K$ basis vectors, and normalize them via a softmax function to obtain weights $w_i$, obtaining the final score as the weighted sum of the $K$ basic scores.

$$w_i = \frac{\exp(\cos < \mathbf{U}^\top \mathcal{E}_{align}(\mathbf{I}), \boldsymbol{\mu}_i >)}{\sum_{j=1}^K \exp(\cos < \mathbf{U}^\top \mathcal{E}_{align}(\mathbf{I}), \boldsymbol{\mu}_j >)}, \quad \hat{f} = \sum_{i=1}^K w_i f_i. \tag{3}$$

During training, we initialize with the PCA and bucketed $k$-means results and continue to finetune them by fitting the predicted scores $\hat{f}$ to the score labels $s$. After end-to-end learning, the basis vectors align better with human preferences and yield more accurate scoring results. During inference, we only need to store the fixed vectors $\boldsymbol{\mu}_i$ and $f_i$, inference reduces to simple dot products with them, incurring minimal computational overhead.

# 5 EXPERIMENTS

## 5.1 EXPERIMENTAL SETUP

**Datasets and Metrics.** Following the setup of Q-Insight (Li et al., 2025), we evaluate on a broad suite of IQA datasets spanning four groups: (a) in-the-wild collections—KonIQ (Hosu et al., 2020), SPAQ (Fang et al., 2020), and LIVE-Wild (Ghadiyaram & Bovik, 2015); (b) synthetic distortions—KADID (Lin et al., 2019), CSIQ (Larson & Chandler, 2010) and TID2013 (Ponomarenko et al., 2015); (c) model-processed distortions—PIPAL (GU et al., 2020); and (d) AI-generated images—AGIQA (Li et al., 2023). Mean Opinion Scores (MOS) from all datasets are rescaled to the interval $[1, 5]$. For score regression, we report performance using the Pearson Linear Correlation Coefficient (PLCC) and the Spearman Rank-Order Correlation Coefficient (SRCC).

**Implementation Details.** For RALI, we choose CLIP-VIT-LARGE-PATCH14-336 as our vision encoder. The learning rate is set to $1 \times 10^{-5}$ in contrastive alignment and set to $3 \times 10^{-2}$ in scoring fitting. We use PCA to reduce the dimension of the original feature space from 768 to 512 to reduce some noise and interference. The number of basic vectors and buckets are respectively set to 250 and 240. For RACT, we use QWEN-2.5-VL-7B-INSTRUCT (Bai et al., 2025) as our baseline models. Within GRPO, we sample $N = 8$ candidates per update and apply a KL regularizer with coefficient $\beta = 1 \times 10^{-3}$. The auxiliary losses use weights $\alpha_1 = 0.25$ and $\alpha_2 = 0.75$. We use AdamW (Loshchilov & Hutter, 2017) with an initial learning rate of $1 \times 10^{-6}$ that decays linearly to $1 \times 10^{-9}$ over training. The model is trained for 10 epochs with a batch size of 128, and the full run completes in approximately one day on 8 NVIDIA H20 GPUs.

## 5.2 RESULTS OF SCORE REGRESSION

**Results of Single-dataset Training with RALI.** To evaluate the effectiveness of our proposed RALI, we compare our method with handcrafted methods such as NIQE (Mittal et al., 2012b); non-MLLM deep-learning methods including NIMA Talebi & Milanfar (2018), MUSIQ (Ke et al., 2021), CLIP-IQA+ (Wang et al., 2023), and ManIQA (Yang et al., 2022); and recent MLLM-based methods such as C2Score (Zhu et al., 2024), Q-Align (Wu et al., 2024b), DeQA-Score (You et al., 2025), supervised fine-tuned Qwen (Bai et al., 2025), and Q-Insight (Li et al., 2025). For a fair comparison, all methods (except handcrafted ones) are trained on the KonIQ dataset.

Our comparison results are reported in Table 2. It can be observed that our method achieves competitive results to the SOTA model Q-Insight on PLCC and SRCC. Meanwhile, compared to Q-Insight (7B parameters), we only use about **4%** of the parameters and significantly shorten the running time and storage overhead. Compared to the SOTA none MLLM-based method CLIP-IQA+, our method also surpasses it by 0.056 and 0.059 on PLCC and SRCC respectively. This fully demonstrates the efficiency of our reasoning-free scheme and the accuracy of score fitting.

**Results of Multi-dataset Co-training with RACT.** RACT's results for cross-domain training are presented in Table 3, with comparisons to four baseline methods: VisualQuality-R1 (Liu et al., 2025c), Q-Align (Wu et al., 2024b), DeQA (You et al., 2025), and Q-Insight (Li et al., 2025). Among these, Q-Align and DeQA are SFT-based MLLMs, while Q-Insight and VisualQuality-R1 are RL-based MLLMs. All algorithms are co-trained on the KonIQ, SPAQ, KADID, and PIPAL datasets, and test results are split into in-domain and out-of-domain groups for detailed comparison.

On in-domain datasets, SFT-based algorithms show a clear advantage over RL-based counterparts. Specifically, Q-Insight exhibits poor in-domain fitting due to the lack of cross-domain alignment; On out-of-domain datasets, RL optimized via RACT achieves the highest performance across all out-of-domain datasets.

Table 2: **PLCC / SRCC comparison on the single-domain score regression tasks** between RALI and other competitive IQA methods. All methods except handcrafted ones are trained on the KonIQ dataset. The best and second-best results of each test setting are highlighted in **bold red** and underlined blue.

| Category | Methods | KonIQ | SPAQ | KADID | PIPAL | LiveW | AGIQA | CSIQ | AVG. |
|---|---|---|---|---|---|---|---|---|---|
| Handcrafted | NIQE | 0.533 | 0.679 | 0.468 | 0.195 | 0.493 | 0.560 | 0.718 | 0.521 |
| | (Mittal et al., 2012b) | /0.530 | /0.664 | /0.405 | /0.161 | /0.449 | /0.533 | /0.628 | /0.481 |
| | BRISQUE | 0.225 | 0.490 | 0.429 | 0.267 | 0.361 | 0.541 | 0.740 | 0.436 |
| | (Mittal et al., 2012a) | /0.226 | /0.406 | /0.356 | /0.232 | /0.313 | /0.497 | /0.556 | /0.369 |
| MLLM-based | C2Score | 0.923 | 0.867 | 0.500 | 0.354 | 0.786 | 0.777 | 0.735 | 0.706 |
| | (Zhu et al., 2024) | /0.910 | /0.860 | /0.453 | /0.342 | /0.772 | /0.671 | /0.705 | /0.673 |
| | Q-Align | 0.941 | 0.886 | 0.674 | 0.403 | 0.853 | 0.772 | 0.671 | 0.705 |
| | (Wu et al., 2024b) | /0.940 | /0.887 | /0.684 | /0.419 | /0.860 | /0.735 | /0.737 | /0.752 |
| | DeQA | 0.953 | 0.895 | 0.694 | 0.472 | 0.892 | 0.809 | 0.787 | 0.786 |
| | (You et al., 2025) | /0.941 | /0.896 | /0.687 | /0.478 | /0.879 | /0.729 | /0.744 | /0.765 |
| | VisualQuality-R1 | 0.923 | 0.891 | 0.712 | 0.441 | 0.874 | 0.822 | 0.712 | 0.768 |
| | (Wu et al., 2025b) | /0.908 | /0.892 | /0.711 | /0.438 | /0.849 | /0.767 | /0.662 | /0.747 |
| | Q-Insight | 0.933 | 0.907 | 0.742 | 0.486 | 0.893 | 0.811 | 0.870 | 0.806 |
| | (Li et al., 2025) | /0.916 | /0.905 | /0.736 | /0.474 | /0.865 | /0.764 | /0.824 | /0.783 |
| Non-MLLM Deep-learning | MUSIQ | 0.924 | 0.868 | 0.575 | 0.431 | 0.789 | 0.722 | 0.771 | 0.726 |
| | (Ke et al., 2021) | /0.929 | /0.863 | /0.556 | /0.431 | /0.830 | /0.630 | /0.710 | /0.707 |
| | ManIQA | 0.895 | 0.864 | 0.654 | 0.419 | 0.805 | 0.685 | 0.719 | 0.720 |
| | (Yang et al., 2022) | /0.849 | /0.768 | /0.499 | /0.457 | /0.849 | /0.723 | /0.623 | /0.681 |
| | CLIP-IQA+ | 0.909 | 0.866 | 0.653 | 0.427 | 0.832 | 0.736 | 0.772 | 0.742 |
| | (Wang et al., 2023) | /0.834 | /0.758 | /0.465 | /0.452 | /0.832 | /0.636 | /0.627 | /0.658 |
| | LIQE | 0.901 | 0.860 | 0.720 | 0.480 | 0.842 | 0.739 | 0.782 | 0.761 |
| | (Zhang et al., 2023) | /0.895 | /0.862 | /0.735 | /0.501 | /0.865 | /0.697 | /0.808 | /0.766 |
| | RALI | 0.939 | 0.897 | 0.723 | 0.527 | 0.896 | 0.779 | 0.828 | 0.798 |
| | (Ours) | /0.922 | /0.897 | /0.725 | /0.528 | /0.876 | /0.715 | /0.788 | /0.779 |

Table 3: **PLCC / SRCC comparison on the cross-domain score regression tasks** between RACT and other MLLMs based IQA methods. All methods are trained on the KonIQ, SPAQ, KADID and PIPAL datasets.

| Methods | In-domain | | | | | Out-of-domain | | | | |
|---|---|---|---|---|---|---|---|---|---|---|
| | KonIQ | SPAQ | KADID | PIPAL | AVG. | LiveW | AGIQA | CSIQ | TID13 | AVG. |
| Q-Align | 0.926 | 0.917 | 0.950 | 0.702 | 0.874 | 0.853 | 0.765 | 0.838 | 0.811 | 0.817 |
| (Wu et al., 2024b) | /0.932 | /0.920 | /0.954 | /0.671 | /0.869 | /0.845 | /0.722 | /0.789 | /0.795 | /0.788 |
| DeQA | 0.958 | 0.932 | 0.963 | 0.724 | 0.894 | 0.877 | 0.770 | 0.863 | 0.828 | 0.835 |
| (You et al., 2025) | /0.946 | /0.929 | /0.961 | /0.690 | /0.882 | /0.857 | /0.735 | /0.807 | /0.796 | /0.799 |
| VisualQuality-R1 | 0.899 | 0.918 | 0.918 | 0.603 | 0.834 | 0.852 | 0.812 | 0.859 | 0.799 | 0.831 |
| (Liu et al., 2025c) | /0.881 | /0.914 | /0.920 | /0.588 | /0.826 | /0.834 | /0.753 | /0.772 | /0.764 | /0.781 |
| Q-Insight | 0.899 | 0.913 | 0.757 | 0.579 | 0.787 | 0.867 | 0.805 | 0.768 | 0.743 | 0.796 |
| (Li et al., 2025) | /0.871 | /0.907 | /0.765 | /0.559 | /0.776 | /0.830 | /0.757 | /0.720 | /0.651 | /0.740 |
| RACT | 0.928 | 0.922 | 0.919 | 0.642 | 0.853 | 0.881 | 0.813 | 0.892 | 0.844 | 0.858 |
| (Ours) | /0.907 | /0.918 | /0.916 | /0.626 | /0.842 | /0.846 | /0.763 | /0.838 | /0.817 | /0.816 |

## 5.3 ABLATION STUDIES

**Ablation on RALI's Key Components.** To assess the effectiveness of each component in RALI, we conduct ablation studies on the key components including Contrastive Alignment, PCA Reduction, Bucketed K-Means, Seed Augmentation, and Scoring Definition. The average PLCC and SRCC results, computed in line with single-domain experimental settings, are presented in Table 4. When contrastive alignment is omitted and the original CLIP weights are used directly, we observe a significant degradation in scoring performance. This is because CLIP primarily attends to high-level semantic space and does not adequately interpret the quality reasoning text. When removing PCA reduction and directly use CLIP's native 768-D features, we observe a slight drop in scoring performance, since PCA effectively removes noise in feature fitting and improves generalization. Replacing bucketed k-means with standard k-means leads to a substantial degradation in RALI's IQA performance, as the resulting cluster-based scores are overly concentrated and fail to cover the full score range. Without using multiple seeds to augment quality reasoning text, the CLIP model

Table 4: **Ablation studies on the key components of RALI.** It can be observed that alignment to descriptions and scoring definition based on basis vectors with scores significantly enhance the performance of our method.

|  | Case 1 | Case 2 | Case 3 | Case 4 | Case 5 | Case 6 |
|---|---|---|---|---|---|---|
| Contrastive Alignment | ✗ | ✓ | ✓ | ✓ | ✓ | ✓ |
| PCA Reduction | ✓ | ✗ | ✓ | ✓ | ✓ | ✓ |
| Bucketed K-Means | ✓ | ✓ | ✗ | ✓ | ✓ | ✓ |
| Seed Augmentation | ✗ | ✓ | ✓ | ✗ | ✓ | ✓ |
| Scoring Definition | ✓ | ✓ | ✓ | ✓ | ✗ | ✓ |
| AVG. PLCC | 0.748 | 0.792 | 0.785 | 0.793 | 0.743 | **0.798** |
| AVG. SRCC | 0.727 | 0.772 | 0.766 | 0.773 | 0.723 | **0.779** |

Table 5: **Ablation on labels and training modules in cross-domain SFT.** Scores yield no out-of-domain gain, and fine-tuning only the Visual Encoder (VE as in the table) suffices for comparable performance, as cross-domain reasoning is aligned.

| # | Label | | Train Module | | In-domain | | Out-of-domain | |
|---|---|---|---|---|---|---|---|---|
| | Text | Score | VE | LLM | KonIQ | KADID | CSIQ | AGIQA |
| 1 | ✓ | ✓ | ✓ | ✓ | 0.926 / 0.903 | 0.924 / 0.920 | 0.878 / 0.843 | 0.804 / 0.752 |
| 2 | ✓ | | ✓ | | 0.927 / 0.905 | 0.920 / 0.915 | 0.883 / 0.820 | 0.808 / 0.751 |
| 3 | ✓ | | ✓ | ✓ | 0.928 / 0.907 | 0.919 / 0.916 | 0.881 / 0.846 | 0.813 / 0.763 |

is insufficiently aligned and struggles to converge well. Finally, even without defining and fitting scores on the dimension-reduced basic vectors, the model already surpasses CLIP-IQA+, which demonstrates the effectiveness of our contrastive alignment; adding the score definition further improves the accuracy of score prediction.

**Ablation on Labels and Training Modules in RACT.** We incorporated scores into cross-domain SFT and found they only benefit in-domain performance, with no out-of-domain improvement. The reason is that cross-dataset annotations carry annotator biases—text retains objective quality, but scores incorporate subjective aspects. Training on scores makes the model overfit these variable biases, hence no out-of-domain gain. As discussed earlier, single-dataset-learned reasoning is generalizable, so cross-dataset training only needs to tune the Visual Encoder for cross-domain image inputs. We conducted comparative experiments, and the results are shown in Table 5. Training the Visual Encoder alone and joint training with the LLM yielded comparable performance, which is consistent with our earlier conclusion. However, we observed slower convergence when training the Visual Encoder only.

## 5.4 EFFICIENCY STUDIES OF RALI

As discussed earlier, reasoning MLLMs consume substantial GPU memory due to large parameters and require multistep reasoning, further raising inference costs. RALI offers strong generalization with drastically lower deployment and inference costs than MLLMs. The tests on the NVIDIA A100 (80GB), as shown in Figure 6, reveal the marked efficiency advantage of RALI over Q-Insight: at batch size 16, it consumes only 14.7% of Q-Insight's memory and 3.4% of its inference time.

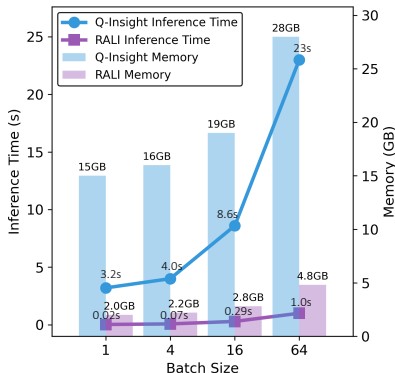

Figure 6: **Efficiency comparison** between Q-Insight and RALI.

## 6 CONCLUSION

In this paper, we revisit reasoning MLLMs in image quality assessment and find their generalization stems from compressing visual information into descriptive text—a compact, domain-bridging representation. Building on this, we pursue two complementary directions. To start with, we leverage this textual representation to develop the Reasoning-Aligned Cross-domain Training (RACT) framework, addressing divergent data distributions: it delivers SOTA out-of-distribution performance on mixed training. Going a step further, we propose the Reasoning-Aligned Lightweight IQA (RALI) framework, which matches reasoning MLLMs in image-to-text mapping by integrating contrastive learning (image-text alignment), PCA (dimensionality reduction), and bucketed K-means (label-text conversion) to delineate quality scoring space. It achieves comparable performance with only 0.3B parameters and no explicit reasoning. Overall, our work reveals how reasoning MLLMs generalize in IQA, provides efficient high-performance solutions, and informs future IQA model design.

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

APPENDIX

# A  THE RELATIONSHIP BETWEEN REASONING AND GENERALIZATION IN Q-INSIGHT

**More Experiments.** To further investigate the relationship between reasoning and generalizability, we have conducted an experiment where Qwen2.5-VL is trained on KonIQ via GRPO without incorporating the reasoning process in Table A.1. Through comparisons, we observe that while GRPO training yields a modest performance improvement over SFT, Q-Insight with the reasoning capability disabled exhibits a significant performance drop, especially in the OOD testing dataset. This validates the correlation between reasoning and generalization in this task.

**More Discussions.** Beyond experiments, we further elaborate on the relationship between reasoning and generalization. The Q-Insight paper already validated this via comparative experiments and analyses of Qwen-SFT and a suite of SFT models. Q-Insight's core motivation and insight is to inspire the large model to "reason deeply and develop insightful perspectives on image quality metrics during scoring" — rather than merely teaching it "how to score images." Removing the reasoning process reduces training to mere score-based constraints, causing the model to forfeit this critical advantage.

Furthermore, many previous works have discussed the relationship between reasoning and generalization. For instance, Visual-RFT (Liu et al., 2025c) also notes that "The reasoning process is key to the model's self-learning and improvement during reinforcement fine-tuning." The essence of Visual-RFT lies in teaching the model "how to think" rather than "what answer to memorize," which constitutes its core competitive advantage over traditional SFT.

In addition, during training Q-Insight, we observed that excessively short reasoning lengths leads to performance drop. Other studies have also discussed related phenomena; for example, (Jin et al., 2024) mentions that insufficient reasoning length can impair LLM performance. Conversely, removing reasoning entirely represents the most extreme case and would result in a significant decline in generalization.

Table A.1: PLCC / SRCC comparison of different training strategies. Q-Insight with the reasoning capability disabled exhibits a significant performance drop.

| Method | KonIQ | SPAQ | KADID | PIPAL | LiveW | AGIQA | CSIQ | AVG. |
|---|---|---|---|---|---|---|---|---|
| Qwen-SFT | 0.889/0.866 | 0.874/0.875 | 0.668/0.663 | 0.473/0.442 | 0.734/0.728 | 0.813/0.739 | 0.674/0.650 | 0.732/0.709 |
| Qwen-GRPO (w/o Reasoning) | 0.921/0.901 | 0.904/0.901 | 0.701/0.698 | 0.459/0.452 | 0.878/0.852 | 0.787/0.728 | 0.728/0.677 | 0.768/0.744 |
| Qwen-GRPO (w/ Reasoning, Q-Insight) | 0.933/0.916 | 0.907/0.905 | 0.742/0.736 | 0.486/0.474 | 0.893/0.865 | 0.811/0.764 | 0.870/0.824 | 0.806/0.783 |

# B  LIMITATIONS AND BROADER IMPACTS

**Limitations.** Although our lightweight RALI achieves strong results, its performance ceiling is still constrained by the representational and reasoning capacity of the underlying CLIP vision encoder. In future work, we will explore stronger CLIP variants (e.g., SigLIP (Zhai et al., 2023)). Meanwhile, following Q-Insight, our experiments primarily target natural-image IQA; however, we believe our reasoning-aligned lightweight approach, together with cross-domain training, can be readily extended to video and AIGC quality assessment.

**Broader Impacts.** Our analysis of reasoning-based MLLMs is not confined to image quality assessment, it readily extends to broader vision–language tasks. In particular, our examination of attention mechanisms in reasoning MLLMs, our exploration of compact textual representation spaces, and our considerations for mitigating cross-domain bias offer actionable insights for future research. Moreover, the proposed reasoning-aligned lightweight IQA framework provides a general and convenient pathway to convert reasoning-based evaluators into reasoning-free ones. This efficient paradigm not only facilitates on-device deployment, but also substantially streamlines the use of reward models in post-training pipelines, such as (Liu et al., 2025b;a).

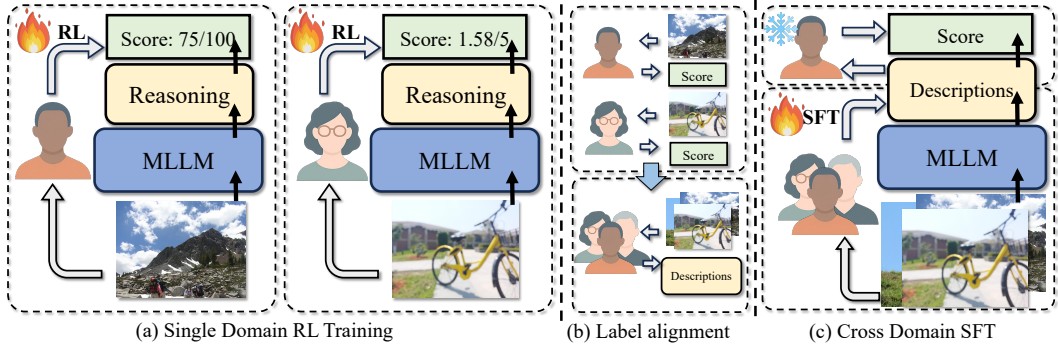

(a) Single Domain RL Training     (b) Label alignment     (c) Cross Domain SFT

Figure A.1: Illustrations of the proposed **R**easoning-**A**ligned **C**ross-Domain **T**raining **F**ramework (RACT). (a) Single-domain RL: We train an MLLM on each IQA dataset to produce reasoning and scores. (b) Label alignment: We use the reasoning module to convert images into quality reasoning text, forming unified image–text labels across datasets. (c) Cross-domain SFT: We finetune the RL-trained model with the aligned image–text pairs to adapt the visual encoder across domains; only one dataset's scores are used to stabilize convergence.

## C   MORE DETAILS ABOUT REASONING-ALIGNED CROSS-DOMAIN TRAINING FRAMEWORK.

**Framework and More Discussion.** A detailed pipeline is illustrated in Figure A.1, the model trained in this manner can be conceptualized as follows: we train the image-to-description conversion module using annotation information from multiple datasets, while the description-to-score prediction module is trained solely on annotations from a single dataset. In our ablation studies, we further note that incorporating scores from multiple datasets into SFT results in degraded performance. Our interpretation of this phenomenon lies in the dual-component nature of dataset annotations: they encompass both objective image quality and annotator group bias. Through reinforcement learning, the model has already acquired objective image quality reasoning text—ones that possess generalizability and domain-bridging capabilities. What remains, however, is annotator group bias, which exhibits a substantial gap across different dataset domains. A more intuitive illustration of this is: if groups A, B, and C are mutually unrelated, fitting group C's preferences using only group A's biases yields no improvement compared to fitting them using the combined biases of groups A and B.

**Advantages of RACT Agianst Naive Multi-dataset RL Training.** To enable a clearer comparison between RACT and Q-Insight, we conducted two additional experiments: specifically, mixed training experiments of both methods on the KonIQ and SPAQ datasets. These experimental results are consolidated in a single table to facilitate comparison.

As shown in Table A.2: 1) Compared to Q-Insight trained on a single dataset, mixed training leads to a performance degradation of Q-Insight, and this degradation becomes more severe as the number of mixed datasets increases. 2) In the same training settings, RACT exhibits better convergence and generalization. Moreover, as more datasets are included, the additional gains of RACT become more significant.

Table A.2: PLCC / SRCC comparison between Q-Insight and RACT across different training settings and datasets.

| Training Datasets | Method | KonIQ | SPAQ | KADID | PIPAL | LiveW | AGIQA | CSIQ | TID | AVG. |
|---|---|---|---|---|---|---|---|---|---|---|
| KonIQ | Q-Insight | 0.933/0.916 | 0.907/0.905 | 0.742/0.736 | 0.486/0.474 | 0.893/0.865 | 0.811/0.764 | 0.870/0.824 | 0.749/0.656 | 0.798/0.767 |
| KonIQ, SPAQ | Q-Insight | 0.928/0.913 | 0.928/0.924 | 0.717/0.711 | 0.496/0.482 | 0.886/0.861 | 0.816/0.747 | 0.816/0.754 | 0.728/0.638 | 0.789/0.754 |
| | RACT | 0.929/0.909 | 0.922/0.918 | 0.752/0.742 | 0.477/0.467 | 0.878/0.847 | 0.814/0.757 | 0.890/0.842 | 0.752/0.662 | 0.802/0.768 |
| KonIQ, SPAQ, KADD, PIPAL | Q-Insight | 0.899/0.871 | 0.913/0.907 | 0.757/0.765 | 0.579/0.559 | 0.867/0.830 | 0.805/0.757 | 0.768/0.720 | 0.743/0.651 | 0.791/0.757 |
| | RACT | 0.928/0.907 | 0.922/0.918 | 0.919/0.916 | 0.642/0.626 | 0.881/0.846 | 0.813/0.763 | 0.892/0.838 | 0.844/0.817 | 0.855/0.829 |

**The Impact of Different Score-labeled Datasets on RACT's Performance.** In the experiments presented in Table 3, only the numerical scores from KonIQ were used to train RACT—specifically, cross-domain training was conducted based on Q-Insight pre-trained on KonIQ. We conducted cross-domain training with RACT using Q-Insight pre-trained on SPAQ and performed comparative anal-

yses, leading to the conclusion that RACT is not sensitive to the dataset from which training numeric scores are sourced.

Table A.3: PLCC / SRCC comparison when training RACT with numeric scores from different datasets.

| Scores From | KonIQ | SPAQ | KADID | PIPAL | LiveW | AGIQA | CSIQ | TID | AVG. |
|---|---|---|---|---|---|---|---|---|---|
| KonIQ | 0.928/0.907 | 0.922/0.918 | 0.919/0.916 | 0.642/0.626 | 0.881/0.846 | 0.813/0.763 | 0.892/0.838 | 0.844/0.817 | 0.858/0.816 |
| SPAQ | 0.917/0.892 | 0.925/0.920 | 0.921/0.919 | 0.630/0.613 | 0.880/0.848 | 0.806/0.761 | 0.897/0.844 | 0.841/0.812 | 0.852/0.826 |

**Designed Prompts.** The prompts designed for each task in RACT are detailed in Tab. A.4. For the single dataset RL training, the input includes a task-specific prompt and the image to be rated, with the Mean Opinion Score (MOS) serving as the ground-truth. For the multi-datasets SFT, the input includes a task-specific prompt and the image to be described, with the quality reasoning text serving as the ground-truth.

Table A.4: **Prompts for RACT.**

| |
|---|
| **System Prompt for RL Training:** A conversation between User and Assistant. The user asks a question, and the Assistant solves it. The assistant first thinks about the reasoning process in the mind and then provides the user with the answer. The reasoning process and answer are enclosed within $< think >< /think >$ and $< answer >< /answer >$ tags, respectively, i.e., $< think >$ reasoning process here $< /think >< answer >$ answer here $< /answer >$. |
| **Prompt for Score Regression Task:** What is your overall rating on the quality of this picture? The rating should be a float between 1 and 5, rounded to two decimal places, with 1 representing very poor quality and 5 representing excellent quality. Return the final answer in JSON format with the following keys: "rating": The score. |
| **Prompts for Quality Description Task:** What is your overall assessment of the quality of this picture? |

## D   MORE ABLATION STUDIES

**Ablation on the Hyperparameters of the Proposed RALI.** To further demonstrate the rationality of our hyperparameter choices, we sweep the PCA dimension, the number of basis vectors, and the number of buckets. Results are reported in Table A.5. We find that when the PCA reduction is too low or the number of basic vectors is too small (e.g., 256-D / 100), both these settings (case (1)-(6)) and 512-D (case (8)) can achieve roughly 0.745 PLCC without score definition. However, due to substantial information loss, the dimension-reduced bases cannot adequately fit the feature space, and thus after applying score definition they fail to reach a higher performance ceiling. When the number of buckets is too small, the basis scores become overly concentrated and cannot effectively cover the full score range. Conversely, with a very high basis dimensionality (e.g., 700-D), the model tends to perform better in in-domain scenarios but exhibits reduced generalization out of domain. Moreover, excessively high dimensionality and a large number of bases increase the optimization difficulty of RALI. Thus, we choose case (8) as our final solution.

## E   RELATION TO MODEL LIGHTWEIGHTING

An alternative strategy for model acceleration involves directly applying conventional model lightweighting techniques to MLLMs. How do these methods compare to our RALI approach? There are distinct differences: Firstly, standard lightweighted algorithms typically remain architecturally homologous to their original counterparts, whereas RALI differs fundamentally from MLLM-based algorithms in terms of architectural design. Secondly, RALI achieves an extreme lightweighting ratio of up to 95% while maintaining a performance comparable to that of MLLMs, an efficiency-performance balance that standard lightweight models do not achieve. To validate this, we conducted a controlled experiment: we performed reinforcement learning training on the 3B Qwen-VL model using the KonIQ dataset, with results presented in Table A.6. Experimental findings reveal that when the parameter count is reduced by approximately 50%, the MLLM exhibits

Table A.5: Ablation study on the hyperparameter selection of RALI.

| Case | Score Definition | PCA Dimension | Basic Vectors | Bucket Bins | PLCC | SRCC |
|------|------------------|---------------|---------------|-------------|------|------|
| 1 | × | 256 | 100 | 90 | 0.748 | 0.720 |
| 2 | ✓ | 256 | 100 | 90 | 0.785 | 0.764 |
| 3 | × | 256 | 250 | 240 | 0.747 | 0.718 |
| 4 | ✓ | 256 | 250 | 240 | 0.783 | 0.762 |
| 5 | × | 512 | 100 | 90 | 0.745 | 0.717 |
| 6 | ✓ | 512 | 100 | 90 | 0.783 | 0.762 |
| 7 | × | 512 | 250 | 240 | 0.743 | 0.723 |
| 8 | ✓ | 512 | 250 | 240 | **0.798** | **0.779** |
| 9 | × | 700 | 250 | 240 | 0.745 | 0.719 |
| 10 | ✓ | 700 | 250 | 240 | 0.787 | 0.767 |

a significant performance drop, and 3B Q-Insight cannot skip the reasoning. This confirms that lightweighting Q-Insight cannot match the performance of RALI.

Table A.6: PLCC / SRCC comparison on single-domain score regression tasks. All methods are trained on the KonIQ dataset. Q-Insight (3B) shows significantly lower performance than RALI (Ours) after 50% parameter reduction.

| Methods | KonIQ | SPAQ | KADID | PIPAL | LiveW | AGIQA | CSIQ | AVG. |
|---------|-------|------|-------|-------|-------|-------|------|------|
| Q-Insight (3B) | 0.907 | 0.897 | 0.704 | 0.445 | 0.824 | 0.831 | 0.826 | 0.776 |
| (Li et al., 2025) | /0.887 | /0.892 | /0.699 | /0.452 | /0.788 | /0.758 | /0.798 | /0.753 |
| RALI | 0.939 | 0.897 | 0.723 | 0.527 | 0.896 | 0.779 | 0.828 | 0.798 |
| (Ours) | /0.922 | /0.897 | /0.725 | /0.528 | /0.876 | /0.715 | /0.788 | /0.779 |

## F  VISUALIZATION

We further present visualization comparisons of reasoning traces and scores between our RACT and VisualQuality-R1 in Figures A.2 and A.3. As shown in Figures A.2 and A.3, our method produces more concise reasoning that is better aligned with image quality, and its predicted scores are consistently more accurate than those of VisualQuality-R1.

## G  LLM USAGE STATEMENT

We used a large language model (LLM) only for minor grammar and phrasing polishes. All technical content, including ideas, experiments, analyses, and discussions, was entirely created by the authors.

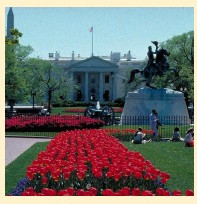

What is your overall rating on the quality of this picture? The rating should be a float between 1 and 5, rounded to two decimal places, with 1 representing very poor quality and 5 representing excellent quality. Return the final answer in JSON format with the following keys: \"rating\": The score.

**GT Score: 3.95**

**<think>** The image appears clear and well-lit, capturing the details of the White House, the statue, and the tulip garden effectively. The colors are vibrant, especially the red tulips, which stand out against the green grass and the white building. There are no visible distortions or blurs that would affect the overall quality. The composition is balanced, with the statue and the White House as focal points. **</think>**
**<answer>** 4.70 **</answer>**

**VisualQuality-R1**

**<think>** The image appears clear with good lighting and sharp details. The colors are vibrant, especially the red tulips, which stand out against the green grass and the white building. There's no visible noise or distortion that would lower the quality significantly. The composition is well-framed, capturing the White House, the statue, and the flowerbeds effectively. **</think>**
**<answer>** 3.92 **</answer>**

**RACT (Ours)**

Figure A.2: Visualization comparison between our proposed RACT and visualquality-R1 (Wu et al., 2025b) on the CSIQ dataset.

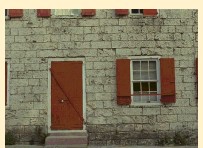

What is your overall rating on the quality of this picture? The rating should be a float between 1 and 5, rounded to two decimal places, with 1 representing very poor quality and 5 representing excellent quality. Return the final answer in JSON format with the following keys: \"rating\": The score.

**GT Score: 3.59**

**<think>** The image appears to be clear and well-lit, with good visibility of the architectural details such as the stone wall texture, the wooden door, and the window with shutters. The colors are natural and there's no noticeable noise or distortion. However, the image seems to have been edited or filtered, which slightly affects its natural appearance but does not significantly degrade the overall quality. **</think>**
**<answer>** 3.80 **</answer>**

**VisualQuality-R1**

**<think>** The image appears clear, with good contrast and sharpness that allows for the details of the stone wall, door, and window to be discernible. There's no evident motion blur or significant distortion affecting the image quality. However, there seems to be some noise or slight graininess present, which might slightly reduce the perceived quality. **</think>**
**<answer>** 3.50 **</answer>**

**RACT (Ours)**

Figure A.3: Visualization comparison between our proposed RACT and visualquality-R1 (Wu et al., 2025b) on the TID2013 dataset.

