# OpenReview forum: "Reasoning as Representation: Rethinking Visual Reinforcement Learning in Image Quality Assessment"
_ICLR.cc/2026/Conference — ICLR 2026 Oral_

### Official Review · Reviewer_6GPX · 2025-10-29

**Soundness:** 3
**Presentation:** 3
**Contribution:** 3
**Rating:** 4
**Confidence:** 5

**Summary:**

This paper investigates the generalization mechanism of reasoning-based reinforcement learning (RL) models for image quality assessment (IQA). The authors argue that the strong generalization of RL-trained multimodal large language models (MLLMs) stems from their ability to compress redundant visual representations into compact, cross-domain text representations through reasoning. Based on this insight, the paper introduces two novel frameworks:

1. RACT (Reasoning-Aligned Cross-domain Training): aligns multiple IQA datasets via reasoning-generated text to improve cross-domain generalization.

2. RALI (Reasoning-Aligned Lightweight IQA): leverages contrastive learning to directly align images with reasoning text embeddings, achieving comparable performance to reasoning-based MLLMs with only 4% of their parameters.

**Strengths:**

1. This paper investigates the generalization of reinforcement learning–based IQA methods. Although I believe the experimental setup is not entirely sufficient, it represents a frontier study in this field and carries significant research value.

2. The RALI framework is quite interesting. Essentially, it distills a lightweight CLIP model using the reasoning text data generated by a reasoning-based model, and it further establishes a mapping between reasoning texts and quality scores. This design effectively avoids the problem of inconsistent score scales across multiple datasets during joint training. Meanwhile, textual information, compared with direct supervision using single numerical scores, may enhance the model’s generalization ability. From this perspective, I think the approach is somewhat similar to that of Q-Align, except that Q-Align directly maps scores to a few fixed textual tokens. I think this work provides an insightful strategy for multi-dataset joint training.

**Weaknesses:**

１.	Line 70–71: The authors state that “Instead of relying on lengthy visual tokens, it now depends on concise and compact reasoning (quality description) text tokens.” I disagree with this statement because, for both reasoning and non-reasoning models, the inputs are images (i.e., visual tokens). The reasoning model additionally generates a group of text tokens during output, derived from score-level supervision through RL. Therefore, it is unclear why the authors claim that reasoning-based models “depend on text tokens.” In my view, those text tokens are outputs rather than inputs.


２.	Line 74–76: The authors state that “previous MLLMs typically predict image quality through visual representations (more than 1000 tokens), whereas reasoning-based models do so via textual representations (less than 100 tokens), resulting in a compression of more than 10 times.” Similar to my comment in Question 1, I do not understand why reasoning-based models are described as not relying on visual tokens. These models still take images as input and thus require visual tokens to generate reasoning text. They cannot directly output reasoning tokens without first processing the visual input. Therefore, I disagree with this statement as currently written.


３.	Figure 2: The caption refers to subfigures (a) and (b), but the figure itself does not contain such labels. Please revise the figure or caption for consistency.


４.	Section 3.2: The comparison between Q-Insight and Qwen-VL appears to be unfair, as Qwen-VL has not been trained on any image quality assessment (IQA) dataset and lacks the capability for quality scoring. It is therefore expected that its reasoning outputs are longer and less relevant than those of Q-Insight, which the t-SNE visualization also reflects. A more appropriate comparison would involve a version of Qwen-VL that has been SFT-trained on the same IQA dataset. Furthermore, I do not fully agree with the conclusion that “through reinforcement learning, the reasoning model shifted its dependency on image quality scoring from visual tokens to reasoning text tokens.” As mentioned in Questions 1 and 2, reasoning text tokens and scores are both outputs of the reasoning model. The text tokens serve to explain why the model assigns a particular score, but the claimed “dependency shift from visual tokens to reasoning text tokens” is conceptually confusing, since these belong to two different stages of the processing pipeline.


５.	Figure 4: The comparison between visual tokens and reasoning text tokens appears to be unfair. To ensure a fair comparison, the visual model should also be fine-tuned on the same dataset, and the visual features used for comparison should be extracted from the last layer of that model to ensure sensitivity to image quality.


６.	Table 1: A compression experiment is missing—specifically, results for Qwen-VL trained via supervised fine-tuning should be added to show its in-domain and out-of-domain performance for comparison.


７.	Line 239: The authors mention that “its PLCC on KonIQ decreased by 0.024 compared to standalone training.” Please provide a reference for this number. Moreover, according to Tables 2 and 3, the proposed model also suffers a 0.011 drop, which should be discussed for completeness.


８.	Line 293–295: The statement “Although high-dimensional visual embeddings can fit the feature space well, they may harm out-of-distribution generalization, so we compress the visual tokens and reduce the visual space.” Please clarify how this conclusion is derived.


９.	Table 2: The order of methods and corresponding references is inconsistent across entries; please standardize the formatting.

10.Tables 2 and 3: An important baseline, LIQE, is missing. LIQE also uses CLIP as its backbone and performs multi-dataset training; it should be included for a fair comparison.

**Questions:**

Please see Weaknesses

---

> ### Author Response · Authors · 2025-11-20
> **Official Comment by Authors**
>
> ## Weakness #1: Text tokens are outputs rather than inputs
> - We kindly remind you that you may have a misunderstandings about our statement. These text tokens are indeed outputs, but due to **the next-token prediction mechanism of large language models**, the final score prediction (i.e. $<$answer$>$3.67$<$/answer$>$) heavily depends on the previously generated reasoning text tokens within the $<$think$>$$<$/think$>$ tags, as claimed in [1, 2]. Our statement is also reflected in Figure 3. We computed the attention between the output token ''\rating'' and all preceding reasoning tokens as well as all visual tokens. We found that the ''\rating'' token attends primarily to the tokens corresponding to the reasoning text instead of the visual tokens, which strongly supports our statement.
>
> ## Weakness #2: About reasoning and scoring mechanism
> - As shown in Figure 3 and Section 3.2, we find that the quality-score token such as "\rating" produced by the reasoning-based LLM has the strongest correlation (highest cross-attention scores) with the reasoning text, and much weaker attention to the visual tokens during the scoring stage.
> - Of course, we acknowledge that the LLM must receive visual tokens to generate the reasoning text in the first place. For rigor and clarity, following your advice, we have revised the sentence in the main paper to: ''Previous MLLMs typically predict image quality through high-dimensional visual representations (more than 1,000 tokens), whereas reasoning-based models rely primarily on the reasoning-text representation during scoring stage (fewer than 100 tokens), resulting in a compression of more than 10 times.''
>
> ## Weakness #3: About Figure 2
> - Thank you for the reminder. For improved readability, partition lines have been added to Figure 2, and labels (a) and (b) have been replaced with (left) and (right).
>
> ## Weakness #4: Comparison between Q-Insight and Qwen-VL
> - This comparison **is not designed to compare or assess the superiority or inferiority of Q-Insight and Qwen-VL**. Qwen-VL inherently exhibits reasoning capabilities, and Q-Insight is fine-tuned based on Qwen-VL. The core purpose of this experiment is to uncover discrepancies in the reasoning process following reinforcement learning (RL) training, thus facilitating a deeper understanding of the relationship between reasoning and the quality scoring task. We find that after RL training, the reasoning becomes more succinct and demonstrates a stronger correlation with quality scores. Additionally, for this specific task, applying SFT to Qwen-VL would impair its intrinsic reasoning ability, rendering such a direct comparison methodologically unfeasible.
>
> ## Weakness #5: Comparison between visual tokens and reasoning text tokens
> - This comparison is based on the fact that both the visual component and the LLM component **undergo the same training process and are trained on the same dataset**. Additionally, the visual tokens used here are the outputs of the visual encoder, i.e., the final layer of visual features.
> - Figure 4 illustrates the domain gap between visual tokens and reasoning text tokens across different datasets (SPAQ and KADID) within the same model (i.e., Q-Insight, trained on KonIQ) during inference. As observed, there exists a significant gap between the features of SPAQ and KADID when represented by visual tokens; however, this domain gap is significantly reduced under the representation of text tokens.
>
> ## Weakness #6: About Table 1
> - **Table 1 is not a comparative experiment but rather an analytical experiment**. Table 1 aims to demonstrate that the IQA reasoning process acquired through RL has a weak correlation with the training datasets, and such a reasoning process is less prone to overfitting to the training data. Table 1 shows that, after isolating the influence of the visual encoder, LLMs trained via RL on different datasets exhibit similar and consistent reasoning processes, while achieving comparable out-of-distribution (OOD) performance. **This comparison itself has no direct connection to SFT**; moreover, SFT disrupts the reasoning process and makes the comparison infeasible. We have revised the table caption to make the meaning of the comparisons in the table more accessible to readers.

---

> ### Author Response · Authors · 2025-11-20
> **Part 2**
>
> ## Weakness #7: RACT's improvement
> - In Table 2, VisualQuality-R1, trained on the single KonIQ dataset and tested on KonIQ, achieves a PLCC of 0.923. In Table 3, VisualQuality-R1, trained on a mixed dataset (KonIQ, SPAQ, KADID, and PIPAL) and tested on KonIQ, yields a PLCC of 0.899. Thus, the degradation caused by mixed training is **0.923 - 0.899 = 0.024**.
> - Since RACT is trained based on the Q-Insight framework, the PLCC performance drop on KonIQ can be derived by comparing the Q-Insight data in Table 3 with the RACT data in Table 2, which is **0.933 - 0.928 = 0.005**. It is evident that the performance drop of RACT is significantly smaller.
> - To facilitate a clearer comparison of RACT's performance against naive mixed RL training, we have supplemented a comparative table, which has been included in the revised appendix.
>
> | Training Datasets | Method | KonIQ | SPAQ | KADID | PIPAL | LiveW | AGIQA | CSIQ | TID | AVG. |
> |-------------------|--------|-------|------|-------|-------|-------|-------|------|-----|------|
> | KonlQ | Q-Insight | 0.933/0.916 | 0.907/0.905 | 0.742/0.736 | 0.486/0.474 | 0.893/0.865 | 0.811/0.764 | 0.870/0.824 | 0.749/0.656 | 0.798/0.767 |
> | KonlQ, SPAQ | Q-Insight | 0.928/0.913 | 0.928/0.924 | 0.717/0.711 | 0.496/0.482 | 0.886/0.861 | 0.816/0.747 | 0.816/0.754 | 0.728/0.638 | 0.789/0.754 |
> | KonlQ, SPAQ | RACT | 0.929/0.909 | 0.922/0.918 | 0.752/0.742 | 0.477/0.467 | 0.878/0.847 | 0.814/0.757 | 0.890/0.842 | 0.752/0.662 | 0.802/0.768 |
> | KonlQ, SPAQ, KADID, PIPAL | Q-Insight | 0.899/0.871 | 0.913/0.907 | 0.757/0.765 | 0.579/0.559 | 0.867/0.830 | 0.805/0.757 | 0.768/0.720 | 0.743/0.651 | 0.791/0.757 |
> | KonlQ, SPAQ, KADID, PIPAL | RACT | 0.928/0.907 | 0.922/0.918 | 0.919/0.916 | 0.642/0.626 | 0.881/0.846 | 0.813/0.763 | 0.892/0.838 | 0.844/0.817 | 0.855/0.829 |
>
> - Under the same training settings, RACT exhibits better convergence and generalization. Moreover, when PIPAL and KADID are included, Q-Insight's mixed training suffers severe convergence issues. In such cases, the RACT approach demonstrates a clear performance advantage, achieving an average PLCC improvement of 0.064 when trained on the mixture of four datasets, **with a notable gain of 0.162 on KADID**.
>
> ## Weakness #8: About dimension compression
> - Compared with tasks such as image classification, IQA tasks tend to rely on generally smaller-scale training datasets, which makes learning-based methods more likely to be susceptible to generalization challenges. With the same level of representational capacity, lower-dimensional representations yield better generalization—excessively high representation dimensions can conversely degrade generalization performance [3]. Thus, we attempt to compress the representation dimension in RALI. This issue is discussed in Section 3.3 of the paper, with relevant citations provided [3]. To enhance readability, we will also include this citation [3] here in the revised version.

---

> > ### Author Response · Authors · 2025-11-28
> >
> > Thanks for your response and for reconsidering the score. We sincerely appreciate the suggestions you provided, which are very helpful for improving our paper. We will incorporate your suggestions into our revised version.

---

> ### Author Response · Authors · 2025-11-20
> **Part3**
>
> ## Weakness #9: Typo in table 2
> - Thank you for the reminder. We have corrected this typo in the revised version.
>
> ## Weakness #10: LIQE's result
> - Thank you for your reminder. Following your advice, we have added the results of LIQE under both the single-dataset training setting using KonIQ and the co-training setting using KonIQ, SPAQ, KADID, and PIPAL. As reported, LIQE shows significantly lower PLCC and SRCC compared to RALI in both settings, far below the performance shown in Tables 2 and 3.
>
> |               | Metrics | KonIQ | SPAQ | KADID | PIPAL | LiveW | AGIQA | CSIQ | AVG. |
> |---------------|---------|-------|------|--------|--------|--------|--------|-------|-------|
> | **LIQE-KonIQ** | PLCC    | 0.901 | 0.860 | 0.720  | 0.480  | 0.842  | 0.739  | 0.782 | 0.761 |
> |               | SRCC    | 0.895 | 0.862 | 0.735  | 0.501  | 0.865  | 0.697  | 0.808 | 0.766 |
> | **RALI**       | PLCC    | 0.939 | 0.897 | 0.723  | 0.527  | 0.896  | 0.779  | 0.828 | 0.798 |
> |               | SRCC    | 0.922 | 0.897 | 0.725  | 0.528  | 0.876  | 0.715  | 0.788 | 0.779 |
>
> |                    | Metrics | KonIQ | SPAQ | KADID | PIPAL | LiveW | AGIQA | CSIQ | TID  | AVG.  |
> |--------------------|---------|--------|--------|--------|--------|--------|--------|--------|--------|--------|
> | **LIQE-Multidataset** | PLCC    | 0.864  | 0.906  | 0.865  | 0.569  | 0.833  | 0.731  | 0.786  | 0.805  | 0.795  |
> |                    | SRCC    | 0.864  | 0.910  | 0.854  | 0.592  | 0.823  | 0.695  | 0.775  | 0.798  | 0.789  |
> | **RACT**           | PLCC    | 0.928  | 0.922  | 0.919  | 0.642  | 0.881  | 0.813  | 0.892  | 0.844  | 0.858  |
> |                    | SRCC    | 0.907  | 0.918  | 0.916  | 0.626  | 0.846  | 0.763  | 0.838  | 0.817  | 0.816  |
>
> ## Reference
> [1] Deepseek-r1: Incentivizing reasoning capability in llms via reinforcement learning.
>
> [2] Ziyu Liu, Zeyi Sun, Yuhang Zang, Xiaoyi Dong, Yuhang Cao, Haodong Duan, Dahua Lin, Jiaqi Wang. Visual-RFT: Visual Reinforcement Fine-Tuning. ICCV 2025.
>
> [3] Yoshua Bengio, Aaron Courville, and Pascal Vincent. Representation learning: A review and new perspectives. IEEE Transactions on Pattern Analysis and Machine Intelligence.

---

> > ### Comment · Reviewer_6GPX · 2025-11-27
> >
> > Thanks for the detailed response. I believe the authors have addressed most of the issues; therefore, I am willing to raise my score.

---

### Official Review · Reviewer_Wmn3 · 2025-10-30

**Soundness:** 3
**Presentation:** 3
**Contribution:** 3
**Rating:** 6
**Confidence:** 4

**Summary:**

The paper investigates the generalization capabilities of reasoning-based IQA models, particularly those trained with reinforcement learning. It argues that these models achieve superior generalization by converting redundant visual representations into compact, cross-domain aligned text representations (quality descriptions). Based on this, the authors propose RALI, a new framework that uses contrastive learning to align images directly with these generalizable text representations, bypassing the need for explicit reasoning and LLMs during inference. The paper demonstrates that RALI achieves comparable performance to reasoning-based models with significantly reduced computational cost.  The authors also introduce RACT, a Reasoning-Aligned Cross-domain Training framework to improve co-training on multiple IQA datasets.

**Strengths:**

1) The core idea of linking generalization in IQA to the compression of visual information into text is insightful.
2) The proposed RALI framework is well-designed and technically sound. The combination of contrastive learning, PCA, and bucketed k-means is effective.
3) The experimental results are compelling and clearly demonstrate the effectiveness of RALI.
4) The paper is generally well-written and easy to follow. The figures and tables are well-designed and informative.

**Weaknesses:**

- While the paper convincingly demonstrates the importance of text representations, it sometimes overstates the case by implying that reasoning itself is not important. Reasoning, even if implicit, is still involved in the process of generating the quality descriptions used to train RALI. The paper needs to acknowledge this nuance.
- RALI relies on Q-Insight to generate the initial image-text-score triplets. This raises concerns about the potential for bias in the generated data. It would be helpful to explore the sensitivity of RALI to the quality of the Q-Insight model.
- The improvement of RACT is not significant enough. The authors should provide more solid evidence to support the effectiveness of RACT.
- Can you clarify the role of reasoning in your framework? While RALI doesn't explicitly perform reasoning during inference, the quality descriptions it relies on are generated by a reasoning-based model. How does the quality of the initial reasoning affect RALI's performance?

**Questions:**

- Have you considered applying RALI to other vision-language tasks, such as image captioning or visual question answering?
- What are the limitations of RALI? What types of images or distortions does it struggle with?
- You mention the reduced computational cost of RALI. Can you provide a more detailed breakdown of the computational cost of each step in the RALI pipeline, compared to Q-Insight?

---

> ### Author Response · Authors · 2025-11-20
>
> ## Weakness #1: RALI's contributions
>
> * Thanks for your reminder. We will clarify the scope of our contributions: RALI can achieve comparable generalization while eliminating the need for a reasoning process and avoiding loading the LLM ​**during inference**​. Meanwhile, we acknowledge that reasoning itself is an important representation and is used to train RALI. We have made this point clearer in the revised version.
>
> ## Weakness #2 & Weakness #4: Role of reasoning and RALI's  sensitivity to the reasoning text
>
> * In our paper, we argue that the reasoning text is a crucial source of cross-domain generalization for LLM-based IQA model. During training, it is used to align the CLIP model with the quality-aware text space optimized via RL, ultimately improving the performance of CLIP-style score-based predictors.
> * **RALI is insensitive to the quality of the generated data and Q-Insight. ​**To verify this, we used Q-Insight to generate image–text–score triplets on the SPAQ training set and used them to align CLIP. Since Q-Insight is trained on KonIQ, the quality of its generated texts slightly degrades when applied to SPAQ. However, we find that RALI-SPAQ still achieves PLCC and SRCC scores of 0.792 and 0.778, which are close to the original RALI results aligned on KonIQ.
>
> | Method      | Metrics | KonIQ | SPAQ | KADID | PIPAL | LiveW | AGIQA | CSIQ | AVG. |
> |-------------|---------|-------|------|-------|-------|-------|-------|------|------|
> | RALL-SPAQ   | PLCC    | 0.939 | 0.897 | 0.721 | 0.491 | 0.903 | 0.763 | 0.833 | 0.792 |
> |             | SRCC    | 0.921 | 0.900 | 0.736 | 0.507 | 0.878 | 0.701 | 0.804 | 0.778 |
> | RALL-KonIQ  | PLCC    | 0.939 | 0.897 | 0.723 | 0.527 | 0.896 | 0.779 | 0.828 | 0.798 |
> |             | SRCC    | 0.922 | 0.897 | 0.725 | 0.528 | 0.876 | 0.715 | 0.788 | 0.779 |
>
>
>
> ## Weakness #3: RACT's Improvement
>
> * Thanks for your reminder. To facilitate clearer comparison, we have supplemented a table and added two additional experiments: specifically, mixed training experiments of Q-Insight and RACT on KonIQ and SPAQ. The table has been added to the revised appendix. As shown in the table:
>
> 1) Compared with Q-Insight trained on a single dataset, mixed training leads to performance degradation of Q-Insight, and this degradation becomes more severe as the number of mixed datasets increases.
> 2) Under the same training settings, RACT exhibits better convergence and generalization. Moreover, as more datasets are included, the additional gains of RACT become more significant. In such cases, the RACT approach demonstrates a clear performance advantage, achieving an average PLCC improvement of 0.064 when trained on the mixture of four datasets, with a ​**notable gain of 0.162 on KADID**​.
>
> | Training Datasets | Method | KonIQ | SPAQ | KADID | PIPAL | LiveW | AGIQA | CSIQ | TID | AVG. |
> |-------------------|--------|-------|------|-------|-------|-------|-------|------|-----|------|
> | KonlQ | Q-Insight | 0.933/0.916 | 0.907/0.905 | 0.742/0.736 | 0.486/0.474 | 0.893/0.865 | 0.811/0.764 | 0.870/0.824 | 0.749/0.656 | 0.798/0.767 |
> | KonlQ, SPAQ | Q-Insight | 0.928/0.913 | 0.928/0.924 | 0.717/0.711 | 0.496/0.482 | 0.886/0.861 | 0.816/0.747 | 0.816/0.754 | 0.728/0.638 | 0.789/0.754 |
> | KonlQ, SPAQ | RACT | 0.929/0.909 | 0.922/0.918 | 0.752/0.742 | 0.477/0.467 | 0.878/0.847 | 0.814/0.757 | 0.890/0.842 | 0.752/0.662 | 0.802/0.768 |
> | KonlQ, SPAQ, KADID, PIPAL | Q-Insight | 0.899/0.871 | 0.913/0.907 | 0.757/0.765 | 0.579/0.559 | 0.867/0.830 | 0.805/0.757 | 0.768/0.720 | 0.743/0.651 | 0.791/0.757 |
> | KonlQ, SPAQ, KADID, PIPAL | RACT | 0.928/0.907 | 0.922/0.918 | 0.919/0.916 | 0.642/0.626 | 0.881/0.846 | 0.813/0.763 | 0.892/0.838 | 0.844/0.817 | 0.855/0.829 |

---

> ### Author Response · Authors · 2025-11-20
> **Part 2**
>
> ## Question #1: More potential applications
>
> * Although RALI is developed within the image quality assessment setting, its overall framework can be extended to other vision tasks such as detection and retrieval.
> * For open-ended tasks like image captioning and visual question answering, RALI’s trained vision tower can be jointly fine-tuned with an LLM to produce texts. We consider this a promising direction for future work.
>
> ## Question #2: Limitations
>
> * We have discussed the limitations of our method in detail in Section A of the supplementary material. The main limitations of RALI are that, although it largely preserves the scoring accuracy of large models, it can only output a score without providing a description. In addition, its performance on unseen AIGC images during training still has room for improvement.
>
> ## Question #3: Computational cost of each steps
>
> * We report below the inference-time breakdown of each component in Q-Insight and RALI. All experiments are conducted on a single H20 GPU with a batch size of 64.
>   | Method    | Tokenizer | Vision Tower | LLM       | Score Mapping | Total Time |
> |-----------|-----------|--------------|-----------|---------------|------------|
> | Q-Insight | 0.5473s   | 1.3921s      | 16.3212s  | None          | 18.2606s   |
> | RALI      | 0.3989s   | 0.9479s      | None      | 0.0009s       | 1.3477s    |
>
>
>
> * Compared with Q-Insight, the efficiency of RALI comes from three aspects. First, RALI does not require the time-consuming LLM reasoning-token generation stage. Second, its vision tower is lighter than that of Q-Insight. Finally, the score-mapping stage in RALI involves only simple PCA projection and cosine-similarity computations, contributing negligible overhead.

---

### Official Review · Reviewer_fd2c · 2025-10-31

**Soundness:** 3
**Presentation:** 3
**Contribution:** 4
**Rating:** 6
**Confidence:** 4

**Summary:**

In this paper, the authors raise two critical questions: how generalization relates to reasoning in image quality assessment (IQA), and whether it is essential, motivated by the observation that IQA scores are largely determined by reasoning text tokens rather than visual tokens.
To answer the first question, the authors propose a reasoning-aligned cross-domain training (RACT) framework to address divergent data distributions by leveraging textual reasoning representations derived from unified cross-dataset labels.
To answer the second question, they propose a reasoning-aligned lightweight IQA (RALI) framework to learn multimodal large language models' (MLLMs) reasoning-based IQA capability by fine-tuning a small vision encoder which is further generalized via feature compression.
In experiments, RALI achieved comparable IQA-scoring performance using only 4% of the parameters and runtime of Q-Insight.

**Strengths:**

In Section 3.2, the paper presents a very interesting observation: the reasoning-based MLLM IQA model, Q-Insight, primarily references text tokens when generating the image quality score token, while hardly referencing visual tokens.
- Building on this observation, the paper's development—proposing a solution to the stated problem—is highly logical and well-grounded.

In particular, by effectively leveraging the relatively domain-invariant nature of text representations compared to visual representations, the method secures generalization performance on the IQA task with a simple approach.
- Both proposed methods, RACT (label alignment and cross-domain SFT) and RALI (feature compression), incorporate solutions for cross-dataset adaptation.
- The approach is validated by the experimental results (Table 2 and Table 3).

Moreover, compared to the prior art Q-Insight, it has a significant advantage for practical deployment by maintaining performance while reducing parameters and runtime to only 4% of Q-Insight.

**Weaknesses:**

Regarding L176: 95% of the attention weights are assigned to the previously generated reasoning text tokens.
- The 95% shown in Figure 3 is for a single example. Reporting the corresponding statistic over the entire KonIQ test set would enhance credibility.
- Performing the same analysis as in Figure 3 on Qwen-VL would help generalize the claim about the reasoning mechanisms of MLLMs in image quality scoring.

Figure 4 may be somewhat self-evident.
- Because KonIQ and SPAQ contain different images, the visual tokens differ substantially, meanwhile, the text space used to describe IQA is relatively constrained (the terms and expressions for IQA are limited within the broader text corpus).
- As in the right plot of Figure 2, adding scores to Figure 4 to show clustering by IQA score regardless of dataset would be more informative.

In the multi-dataset co-training results (Table 3), to better highlight the effect of RACT, it would be preferable to use a combination of four datasets that are very different in nature as described at L335 (e.g., KonIQ, KADID, PIPAL, and AGIQA, rather than KonIQ, SPAQ, KADID, and PIPAL) if possible.

**Questions:**

No questions.

---

> ### Author Response · Authors · 2025-11-20
>
> ## Weakness #1: About attention weights
>
> * We have compiled the overall test results on KonIQ. After excluding the influence of prompts, **93.3%** of the referenced attention is directed toward text tokens when generating score tokens.
>
> * Since Qwen-VL has not been trained with reinforcement learning, it cannot output content in the standard score format of Q-Insight, making it impossible to compute statistics on scores and attention. Furthermore, our main claim is that, compared to SFT models that directly output scores (e.g., DeQA), the attention of score tokens shifts from visual tokens to text tokens, along with changes in reasoning-related attributes such as length. We do not claim any alteration in the inherent pattern of reasoning dependence specific to Qwen-VL itself.
>
> ## Weakness #2: About Figure 4
>
> * Regarding Figure 4, it focuses on the conversion of visual features to textual features in Q-Insight's reasoning. Its core purpose is to demonstrate that the text space used to describe IQA is relatively constrained — precisely the point the figure aims to illustrate. Compared with visual tokens, textual descriptions for image quality are confined to a smaller and more concentrated domain, which is a key advantage of text-based quality representation. Visual features are high-dimensional and complex, while textual features are low-dimensional and compact, enabling good generalization when representing quality.
>
> * Figure 2 differs from Figure 4 in focus: it verifies that after Visual RL optimization, the correlation between reasoning text and quality scores is enhanced. Figure 4, as a 2D t-SNE visualization, suffers from the limitations of low dimensionality — while suitable for revealing domain differences, it inherently fails to depict the correlation with quality scores.
>
> ## Weakness #3: RACT's Effect
>
> * We adopt the experimental protocol of DeQA [1], including its discussion on multi-dataset mixed training. Since DeQA uses the AGIQA dataset exclusively for testing, we consequently omit it from our mixed-training setup. To enable clearer comparison to show RACT's effect, we have supplemented a table and added two additional experiments: specifically, mixed training experiments of Q-Insight and RACT on KonIQ and SPAQ. As shown in the table:
>
> 1. Compared with Q-Insight trained on a single dataset, mixed training causes performance degradation in Q-Insight, with this degradation worsening as the number of mixed datasets increases.
> 2. Under the same training settings, RACT exhibits better convergence and generalization. Moreover, when PIPAL and KADID are included, Q-Insight's mixed training suffers severe convergence issues — this indirectly confirms that PIPAL and KADID differ significantly from KonIQ and SPAQ. In such cases, the RACT approach demonstrates a clear performance advantage, achieving an average PLCC improvement of 0.064 when trained on the mixture of four datasets, with a ​**notable gain of 0.162 on KADID**​.
>
> | Training Datasets | Method | KonIQ | SPAQ | KADID | PIPAL | LiveW | AGIQA | CSIQ | TID | AVG. |
> |-------------------|--------|-------|------|-------|-------|-------|-------|------|-----|------|
> | KonlQ | Q-Insight | 0.933/0.916 | 0.907/0.905 | 0.742/0.736 | 0.486/0.474 | 0.893/0.865 | 0.811/0.764 | 0.870/0.824 | 0.749/0.656 | 0.798/0.767 |
> | KonlQ, SPAQ | Q-Insight | 0.928/0.913 | 0.928/0.924 | 0.717/0.711 | 0.496/0.482 | 0.886/0.861 | 0.816/0.747 | 0.816/0.754 | 0.728/0.638 | 0.789/0.754 |
> | KonlQ, SPAQ | RACT | 0.929/0.909 | 0.922/0.918 | 0.752/0.742 | 0.477/0.467 | 0.878/0.847 | 0.814/0.757 | 0.890/0.842 | 0.752/0.662 | 0.802/0.768 |
> | KonlQ, SPAQ, KADID, PIPAL | Q-Insight | 0.899/0.871 | 0.913/0.907 | 0.757/0.765 | 0.579/0.559 | 0.867/0.830 | 0.805/0.757 | 0.768/0.720 | 0.743/0.651 | 0.791/0.757 |
> | KonlQ, SPAQ, KADID, PIPAL | RACT | 0.928/0.907 | 0.922/0.918 | 0.919/0.916 | 0.642/0.626 | 0.881/0.846 | 0.813/0.763 | 0.892/0.838 | 0.844/0.817 | 0.855/0.829 |
>
> [1] Zhiyuan You, Xin Cai, Jinjin Gu, Tianfan Xue, Chao Dong. Teaching Large Language Models to Regress Accurate Image Quality Scores using Score Distribution. CVPR 2025.

---

### Official Review · Reviewer_xqw8 · 2025-10-31

**Soundness:** 3
**Presentation:** 3
**Contribution:** 2
**Rating:** 4
**Confidence:** 5

**Summary:**

This paper argues that RL-trained reasoning-style IQA models, such as the Q-Insight, don’t generalize better because they’re “better models,” but because RL teaches them to compress rich visual features into short textual descriptions of quality. This text space is said to be both compact and more cross-domain aligned, and that is claimed to be the source of generalization of RL-based IQA models. Based on that, they introduce RACT, Reasoning-Aligned Cross-domain Training, which first does per-dataset RL to obtain quality descriptions, aligns those descriptions across datasets, and then does cross-domain SFT using those aligned image–text pairs. They claimed that the proposed RACT enables RL-style OOD robustness without instability when mixing datasets. Furthermore, they propose RALI, which discards step-by-step reasoning at inference, aligns a CLIP-style vision encoder to those descriptions via contrastive learning, compresses features via PCA + bucketed k-means, and then predicts scores via similarity to a learned bank of basis vectors.

**Strengths:**

1. The paper has a clear attempt to answer new open questions, which is more valuable and challenging than reporting new performance numbers. It tries to explain why RL-style IQA models outperform SFT baselines in cross-dataset generalization, which is timely and valuable regarding the internal mechanism.

2. The proposed RACT and RALI pipelines are conceptually neat and practically actionable. Their effectiveness is verified on diversified IQA datasets. RACT is empirically strong OOD, where it outperforms both SFT-style and RL-style baselines in average PLCC/SRCC, with only the quality descriptions as a unifying label space across different datasets.

**Weaknesses:**

1. The paper repeatedly asserts that RL teaches the model to compress visual tokens into textual reasoning, and that “this conversion is precisely the source of the generalization.” Although interesting, all supporting evidence is correlational, not causal. For example, fig. 3 shows attention mass shifting from visual tokens to reasoning tokens when outputting the score. But this only proves where the model is looking at inference time after reasoning text exists, not that the reasoning step itself is required to learn generalizable features. Provide causal ablations that can quantitatively prove this claim, or this would be fatal for this paper.

2. In section 4, the authors explicitly claim that the proposed RALI eliminates the reliance on reasoning processes and even obviates the need to load an LLM. However, it is also stated that RALI first uses a pre-trained RL IQA model to generate reasoning text between \<think\> and \<think\> and scores for each training image. Therefore, although RALI is cheap at inference, it cannot be trained or updated without first running an RL reasoning model like Q-Insight to produce the textual descriptions that define that space. Can I interpret the contribution in this way: this paper is actually proposing distillation (or compression) of an RL reasoning model into a CLIP-like scorer, instead of removing reasoning entirely.

3. The empirical evidence for RACT’s cross-domain convergence fix is not complete enough in the main part. As claimed in Sec. 3.4, RL on mixed datasets shows a severe convergence problem; the VisualQuality-R1’s ranking-based RL mitigates this but still drops when jointly trained, and multi-dataset reward construction is unstable. However, the paper doesn't quantitatively show the alleged instability/failure modes of naive multi-dataset RL training that motivate RACT. Since RACT is the core contribution, this point is very important but missing.

4. RALI intentionally discards reasoning at inference. That’s great for speed, but some downstream pipelines want textual justifications. The paper doesn’t quantify how often RALI’s score disagrees with the RL model’s score on failure cases where the RL model also gave a plausible, human-auditable explanation.

5. Some of the parameters look tricky and potentially fragile. The score range [1,5] is divided into N=240 buckets, then bucketed k-means with kn clusters per bucket yields 250 basis vectors, and why? Is that engineered around KonIQ’s and Q-Insight’s output distribution?

6. The paper uses “reasoning,” “quality descriptions,” and “text tokens” almost interchangeably. But some of the provided examples look less like multi-step logical reasoning and more like structured perceptual justifications. Calling that “reasoning” may oversell what RL actually learned. I think clearer terminology would help.

In summary, I highly value the motivation and the ambition to answer open questions of the IQA field in this paper. However, I think current experiments and argumentation are not enough to support both the motivation and contribution. Several core claims are currently stronger than the presented evidence supports, especially the causal claim that “reasoning-as-compression is the source of generalization,” and the claim that RALI “removes” reasoning despite relying on RL-generated reasoning text for supervision.

**Questions:**

1. As claimed in the paper, RALI is trained only on KonIQ, and its cross-dataset performance is reported (indicated by Table 2). Are the reasoning texts generated only from KonIQ images?

2. Continuing the previous question, if so, then RALI’s OOD performance hinges on how well Q-Insight’s KonIQ-derived descriptions generalize to distortions and semantic content in SPAQ, AGIQA, CSIQ, etc. But the paper argues earlier that cross-domain alignment of reasoning is what generalizes, which implies we might get even better generalization if we harvest reasoning texts from multiple datasets. However, RALI is not shown in that stronger setting. How would the authors explain this?

3. Which single dataset’s numeric scores are actually used during cross-domain SFT in Table 3’s RACT row? KonIQ only? SPAQ only? How sensitive is RACT to that choice?

---

> ### Author Response · Authors · 2025-11-20
>
> ## Weakness #1: About connection between reasoning and generalization
>
> * We have supplemented an experiment where Qwen2.5-VL is trained on KonIQ via GRPO without incorporating the reasoning process. Through comparisons, we observe that while GRPO training yields a modest performance improvement over SFT, ​**Q-Insight with the reasoning capability disabled exhibits a significant performance drop**​, ​**especially in the OOD testing dataset**​. This validates the correlation between reasoning and generalization in this task.
>
> | Method | KonlQ | SPAQ | KADID | PIPAL | LiveW | AGIQA | CSIQ | AVG. |
> |--------|-------|------|-------|-------|-------|-------|------|------|
> | Qwen-SFT | 0.889/0.866 | 0.874/0.875 | 0.668/0.663 | 0.473/0.442 | 0.734/0.728 | 0.813/0.739 | 0.674/0.650 | 0.732/0.709 |
> | Qwen-GRPO without Reasoning | 0.921/0.901 | 0.904/0.901 | 0.701/0.698 | 0.459/0.452 | 0.878/0.852 | 0.787/0.728 | 0.728/0.677 | 0.768/0.744 |
> | Qwen-GRPO with Reasoning (Q-Insight) | 0.933/0.916 | 0.907/0.905 | 0.742/0.736 | 0.486/0.474 | 0.893/0.865 | 0.811/0.764 | 0.870/0.824 | 0.806/0.783 |
>
> * Beyond experiments, we further elaborate on the relationship between reasoning and generalization. The Q-Insight paper already validated this via comparative experiments and analyses of Qwen-SFT and a suite of SFT models, as reported in Table 2 of the main paper. Q-Insight’s core motivation and insight is to inspire the large model to “*reason deeply and develop insightful perspectives on image quality metrics during scoring*” — rather than merely teaching it “*how to score images*”. Removing the reasoning process reduces training to mere score-based constraints, causing the model to forfeit this critical advantage.
> * Previous works have also discussed how reasoning relates to generalization. For example, Visual-RFT[1] notes that *"The reasoning process is key to the model’s self-learning and improvement during reinforcement fine-tuning."* Its essence lies in teaching *"how to think"* rather than memorizing answers, forming its key advantage over traditional SFT.
> * We also observed during training that overly short reasoning lengths hurt the performance of Q-Insight. Other studies report similar phenomena; [2] mentions that insufficient reasoning impairs LLM performance. Removing reasoning entirely is the extreme case, leading to a decline in generalization.
>
> [1] Ziyu Liu, Zeyi Sun, Yuhang Zang, Xiaoyi Dong, Yuhang Cao, Haodong Duan, Dahua Lin, Jiaqi Wang. Visual-RFT: Visual Reinforcement Fine-Tuning. ICCV 2025.
>
> [2] Mingyu Jin, Qinkai Yu, Dong Shu, Haiyan Zhao. The Impact of Reasoning Step Length on Large Language Models. ACL 2024.
>
> ## Weakness #2: RALI's contribution
>
> * Thank you for your comments. We will clarify the scope of our contributions: RALI can achieve comparable generalization while eliminating the need for a reasoning process and avoiding loading the LLM ​**during inference**​.
> * Regarding the contribution of RALI in the paper, it is a novel image quality scoring method. While its results share similarities with those of distillation and compression techniques, ​**it differs fundamentally in principle**​. In distillation and model compression, the inference-focused student model and the teacher model are generally ​consistent in architecture​. As we compare in Appendix Section D, reducing model parameters based on the original architecture leads to a significant degradation in performance.
> * In contrast, RALI’s architecture is entirely distinct from that of MLLMs. Beyond abandoning LLM and reasoning by directly aligning reasoning text features through contrastive learning during inference, we also leverage text descriptions: we define the image quality scoring space using Bucketed K-means and PCA, and obtain quality scores through end-to-end training within this space.

---

> ### Author Response · Authors · 2025-11-20
> **Part 2**
>
> ## Weakness #3: The divergence problem of naive multi-dataset RL training and the advantages of RACT
>
> * Since RACT is trained on the foundation of Q-Insight, the comparison between Q-Insight (with mixed training) and RACT in Table 2 directly reflects the gains brought by RACT. Additionally, by comparing the performance of Q-Insight (with single-dataset training) in Table 3 with that of RACT in Table 2, the instability of mixed training can be observed.
> * To enable a clearer comparison, we have supplemented a PLCC/SRCC comparison table and added two additional experiments: specifically, mixed training experiments of Q-Insight and RACT on KonIQ and SPAQ. The table has been added to the revised appendix. As shown in the table:
>
> 1) Compared with Q-Insight trained on a single dataset, mixed training leads to performance degradation of Q-Insight, and this degradation becomes more severe as the number of mixed datasets increases.
> 2) Under the same training settings, RACT exhibits better convergence and generalization. Moreover, as more datasets are included, the additional gains of RACT become more significant. In such cases, the RACT approach demonstrates a clear performance advantage, achieving an average PLCC improvement of 0.064 when trained on the mixture of four datasets, with a ​**notable gain of 0.162 on KADID**​.
>
> | Training Datasets | Method | KonIQ | SPAQ | KADID | PIPAL | LiveW | AGIQA | CSIQ | TID | AVG. |
> |-------------------|--------|-------|------|-------|-------|-------|-------|------|-----|------|
> | KonlQ | Q-Insight | 0.933/0.916 | 0.907/0.905 | 0.742/0.736 | 0.486/0.474 | 0.893/0.865 | 0.811/0.764 | 0.870/0.824 | 0.749/0.656 | 0.798/0.767 |
> | KonlQ, SPAQ | Q-Insight | 0.928/0.913 | 0.928/0.924 | 0.717/0.711 | 0.496/0.482 | 0.886/0.861 | 0.816/0.747 | 0.816/0.754 | 0.728/0.638 | 0.789/0.754 |
> | KonlQ, SPAQ | RACT | 0.929/0.909 | 0.922/0.918 | 0.752/0.742 | 0.477/0.467 | 0.878/0.847 | 0.814/0.757 | 0.890/0.842 | 0.752/0.662 | 0.802/0.768 |
> | KonlQ, SPAQ, KADID, PIPAL | Q-Insight | 0.899/0.871 | 0.913/0.907 | 0.757/0.765 | 0.579/0.559 | 0.867/0.830 | 0.805/0.757 | 0.768/0.720 | 0.743/0.651 | 0.791/0.757 |
> | KonlQ, SPAQ, KADID, PIPAL | RACT | 0.928/0.907 | 0.922/0.918 | 0.919/0.916 | 0.642/0.626 | 0.881/0.846 | 0.813/0.763 | 0.892/0.838 | 0.844/0.817 | 0.855/0.829 |
>
> ## ​Weakness #​4: Correlation between RALI's and Q-Insight's predicted scores
>
> * As noted in the paper, RALI is primarily tailored for scenarios demanding real-time performance, limited deployment computing power, and scenarios requiring only scores—such as live streaming quality monitoring and real-time reinforcement learning reward calculation.  Additionally, we conducted an additional quantitative consistency experiment, using Q-Insight’s quality scores as ground truth (GT) to calculate the PLCC and SRCC of RALI’s predicted scores against Q-Insight’s. The experimental results demonstrate that RALI’s output scores **exhibit a high correlation with those from Q-Insight**.
>
> | Metric | KonIQ | SPAQ | KADID |
> |--------|-------|------|-------|
> | PLCC | 0.9452 | 0.9577 | 0.9250 |
> | SRCC | 0.9264 | 0.9480 | 0.9090 |
>
> ## Weakness #5: Parameters selection of RALI:
>
> * Our parameter selection is obtained by sweeping over the PCA dimensionality, the number of basic vectors, and the number of bucket bins, and then choosing the configuration with the highest PCA explainable variance ratio. This process is independent of the output distributions of KonIQ and Q-Insight.
> * We have already provided the hyperparameter sensitivity analysis in Appendix Table A.4, and we include additional analysis in the table below. As reported, our model is insensitive to the choices of bucket bins and basic vectors, consistently achieving around 0.798 PLCC across different cases.
>
> | Case | PCA Dim | Basic vectors | Bucket Bins | PLCC | SRCC |
> |------|---------|---------------|-------------|------|------|
> | (a) | 512 | 250 | 240 | 0.798 | 0.779 |
> | (b) | 512 | 250 | 250 | 0.797 | 0.777 |
> | (c) | 512 | 100 | 90 | 0.796 | 0.776 |
> | (d) | 512 | 100 | 100 | 0.795 | 0.774 |
>
> ## Weakness #6: Terminology
>
> * "Reasoning" is a commonly used term in Reasoning MLLMs. We merely adopt the terminology employed in previous works[1], using "reasoning" to denote the intermediate thinking content during the model’s inference process. To avoid confusion, we have refined the terminology as **"quality reasoning text"** in the revised version.
>
> [1] Ziyu Liu, Zeyi Sun, Yuhang Zang, Xiaoyi Dong, Yuhang Cao, Haodong Duan, Dahua Lin, Jiaqi Wang. Visual-RFT: Visual Reinforcement Fine-Tuning. ICCV 2025.

---

> ### Author Response · Authors · 2025-11-20
> **Part 3**
>
> ## Question #1: RALI's training reasoning texts
>
> * Our reasoning texts are generated by applying the publicly available Q-Insight weights to the KonIQ images, without mixing in any other datasets such as SPAQ, KADID, or PIPAL.
>
> ## Question #2: Potential stronger setting for RALI
>
> * Following your suggestion, we generated reasoning texts using the publicly available Q-Insight on the training sets of KonIQ, SPAQ, KADID, and PIPAL, and aligned CLIP on this combined data (denoted as "multi-dataset"). The resulting average PLCC and SRCC (0.795 and 0.775) are slightly lower than our original setting that uses only KonIQ.
> * We attribute this to the fact that Q-Insight is mainly trained on KonIQ, producing more accurate degradation reasoning on KonIQ images. Its less reliable reasoning on other datasets introduces noise, which in turn weakens the alignment of the CLIP.
>
> | Alignment Datasets | Metrics | KonIQ | SPAQ | KADID | PIPAL | LiveW | AGIQA | CSIQ | AVG. |
> |-------------------|---------|-------|------|-------|-------|-------|-------|------|------|
> | Multi-dataset | PLCC | 0.933 | 0.895 | 0.676 | 0.565 | 0.899 | 0.783 | 0.811 | 0.795 |
> |  | SRCC | 0.917 | 0.893 | 0.699 | 0.570 | 0.875 | 0.711 | 0.758 | 0.775 |
> | KonIQ | PLCC | 0.939 | 0.897 | 0.723 | 0.527 | 0.896 | 0.779 | 0.828 | 0.798 |
> | | SRCC | 0.922 | 0.897 | 0.725 | 0.528 | 0.876 | 0.715 | 0.788 | 0.779 |
>
>
>
> ## Question #3: RACT's used score
>
> * In the experiments presented in Table 3, only the numerical scores from KonIQ were used to train RACT—specifically, cross-domain training was conducted based on Q-Insight pre-trained on KonIQ. We have supplemented cross-domain experimental results using Q-Insight pre-trained on SPAQ and performed comparative analyses, leading to the conclusion that ​**RACT is not sensitive to the dataset from which training numeric scores are sourced**​.
>
> | Scores From | Metrics | KonIQ | SPAQ | KADID | PIPAL | LiveW | AGIQA | CSIQ | TID | AVG. |
> |--------------|---------|-------|------|-------|-------|-------|-------|------|-------|------|
> | KonIQ | PLCC | 0.928 | 0.922 | 0.919 | 0.642 | 0.881 | 0.813 | 0.892 | 0.844 | 0.858 |
> | | SRCC | 0.907 | 0.918 | 0.916 | 0.626 | 0.846 | 0.763 | 0.838 | 0.817 | 0.816 |
> | SPAQ | PLCC | 0.917 | 0.925 | 0.921 | 0.630 | 0.880 | 0.806 | 0.897 | 0.841 | 0.852 |
> | | SRCC | 0.892 | 0.920 | 0.919 | 0.613 | 0.848 | 0.761 | 0.844 | 0.812 | 0.826 |
>
>
> * In Appendix Section B, we have already discussed that incorporating scores from multiple datasets does not yield improvements; similarly, using scores from different individual datasets results in no significant differences. A more intuitive illustration of this is as follows: if groups of people—A, B; and C, D, E, F...—are mutually unrelated, modeling the preferences of groups C, D, E, and F using group A’s biases does not differ significantly from using group B’s biases. We will include this supplementary table in the revised version of the paper.

---

> > ### Comment · Reviewer_xqw8 · 2025-11-28
> > **Great responses.**
> >
> > Thanks for the detailed response from the authors. I think most of the concerns are well addressed. I tend to raise my score to 6 based on the current rebuttal, and I would raise it to 8 if the first point could be further made clear in the paper. But I think the score revision seems to be banned by the system currently. This comment can represent my updated rating if the technical issue cannot be further settled. Thanks.

---

> > > ### Author Response · Authors · 2025-11-28
> > >
> > > We sincerely thank the reviewer for the constructive feedback. Your comments have helped us enhance the quality of the paper. **As recommended, we have now incorporated a discussion on the relationship between reasoning and generalizability in Q-Insight into Section 3.1 to strengthen the logical flow.** The supporting experiments and extended analysis have been moved to Appendix A. We hope that these revisions offer a clearer presentation of the first point regarding Weakness #1. **All changes have been highlighted in green in the revised manuscript**, and we welcome any further suggestions you may have.

---

### Author Response · Authors · 2025-11-20
**Global Response**

Thank you to all the reviewers for their constructive feedback. All reviewers have acknowledged and praised our innovation, methodology, and experimental results. We have addressed all the concerns, responded to each point, and made revisions to the paper. The modified sections are marked in green. We sincerely hope you recheck and reconsider your decision.

---

### Author Response · Authors · 2025-11-27
**Looking Forward to Further Discussions**

Dear reviewers:

We appreciate the time you dedicated to reviewing our work and your recognition of our work. Regarding the concerns you raised, we have provided explanations in our responses. We would like to ensure that your concerns have been adequately addressed. If there are any aspects of our work that remain unclear to you, please don't hesitate to let us know.

Best regards,

Authors of #109

---

### Author Response · Authors · 2025-12-02
**Summary**

Dear Area Chair,

We deeply appreciate your willingness to dedicate your valuable time to conducting a thorough review of our submission and rebuttal amidst the unforeseen circumstances.

To assist your evaluation, we provide a brief summary of our work.

**Paper Overview**

Reasoning MLLM-IQA exhibits strong generalization. To investigate why, we examine its source through two questions: How does reasoning relate to generalization in IQA, and is it essential?

We find generalization stems from quality reasoning text, which condenses redundant visual tokens into compact, cross-domain-aligned text tokens. Based on this, we propose RACT for cross-domain training and shows strong OOD performance. To assess necessity, we introduce RALI. It aligns CLIP with reasoning text via contrastive learning, then defines a score space with bucketed K-means and PCA to enable inference without the LLM. This retains strong generalization while cutting 95% of parameters and inference time, making it suitable for real-time or score-only applications.

**Initial Ratings: ​**4(xqw8), 6(fd2c), 6(Wmn3), 4(6GPX)

**Existing Discussions with Reviewers**

* **Reviewer xqw8**
  * **Great responses.​**​ ​​"Thanks for the detailed response from the authors. I think most of the concerns are well addressed. I tend to raise my score to 6 based on the current rebuttal, and I would raise it to 8 if the first point could be further made clear in the paper. But I think the score revision seems to be banned by the system currently. This comment can represent my updated rating if the technical issue cannot be further settled. Thanks."
* **Reviewer 6GPX**
  * "Thanks for the detailed response. I believe the authors have addressed most of the issues; therefore, I am willing to raise my score."

We regret that the other Reviewers (fd2c, Wmn3) have not yet responded to our rebuttal. However, we have provided detailed responses and conducted extensive new experiments to fully resolve their initial concerns. The responses from the reviewers who originally held negative views indicate that their **concerns have been resolved**. Based on the current state of the discussion, **all reviewers now hold positive views of the paper.**

**Recognized Strengths by All Reviewers**

* Novelty: The paper provides novel insights carrying substantial value for the IQA field.
* Performance: Provided experiments confirm the strong performance of the proposed RALI and RACT.

**Main Concerns and Questions Resolved**

| Issue | Reviewer | Solution |
| :--- | :--- | :--- |
| Relationship between reasoning and generalization | xqw8 | The relationship between reasoning and generalization is further evidenced by additional explanatory analysis and experimental validation in section A of the appendix. |
| RALI's Contribution | xqw8, Wmn3 | We clarify that RALI's contribution is achieving comparable generalization without the reasoning process or the LLM during inference. |
| RACT's Improvement | xqw8, fd2c, Wmn3, 6GPX | A new table A.2 in the appendix provides a clearer comparison of the gains achieved by RACT over naive cross-domain RL training. |
| More Experiments | xqw8, Wmn3, 6GPX | **(1)** Further experiments demonstrate that RALI is robust to a range of design choices, including the number of bucket bins, basis vectors, and the generated reasoning text data. **(2)** Additional experiments demonstrate that RACT is insensitive to the choice of score datasets, and an explanation is provided. **(3)** Added comparisons with LIQE confirm the clear advantage of our proposed method. |


We sincerely thank all reviewers for their constructive feedback. We believe that our revisions, supplementary experiments, and detailed responses have meticulously addressed all concerns. We kindly request that our rebuttal be fully considered in your final decision, and we are truly grateful for your time and expertise throughout this review process.

Best regards,

Authors of #109

---

### Meta-Review · Area_Chair_uVXW · 2026-01-07

**Summary:**

This paper receives 4 high-quality reviews, with 2 positive initial ratings (6) and 2 negative initial ratings (4).

The 2 negative reviewers have participated in the rebuttal discussions, and their concerns have been addressed and they are willing to raise their ratings to positive scores.

The 2 positive reviewers have not participated in the rebuttal discussions, but the authors have provided responses and added new experiments to address the concerns.

**Reviewer Concerns:**

Reviewer xqw8 (initial rating: 4) and Reviewer 6GPX (initial rating: 4)'s concerns have been addressed, and they are willing to raise their ratings.

Reviewer fd2c (initial rating: 6) and Reviewer Wmn3 (initial rating: 6) have not participated in the discussions, but the authors have given good responses.

**Reviewer Scores:**

All reviewers hold positive reviews of the paper after rebuttal.

---

### Decision · Program_Chairs · 2026-01-26

Accept (Oral)